# Two opposite voltage-dependent currents control the unusual early development pattern of embryonic Renshaw cell electrical activity

Juliette Boeri[1†], Claude Meunier[2†], Hervé Le Corronc[1,3†], Pascal Branchereau[4], Yulia Timofeeva[5,6], François-Xavier Lejeune[7], Christine Mouffle[1], Hervé Arulkandarajah[1], Jean Marie Mangin[1], Pascal Legendre[1‡*], Antonny Czarnecki[1,4‡*]

[1]INSERM, UMR_S 1130, CNRS, UMR 8246, Neuroscience Paris Seine, Institute of Biology Paris Seine, Sorbonne Univ, Paris, France; [2]Centre de Neurosciences Intégratives et Cognition, CNRS UMR 8002, Institut Neurosciences et Cognition, Université de Paris, Paris, France; [3]Univ Angers, Angers, France; [4]Univ. Bordeaux, CNRS, EPHE, INCIA, Bordeaux, France; [5]Department of Computer Science and Centre for Complexity Science, University of Warwick, Coventry, United Kingdom; [6]Department of Clinical and Experimental Epilepsy, UCL Queen Square Institute of Neurology, University College London, London, United Kingdom; [7]Institut du Cerveau et de la Moelle Epinière, Centre de Recherche CHU Pitié-Salpétrière, INSERM, U975, CNRS, UMR 7225, Sorbonne Univ, Paris, France

*For correspondence:
pascal.legendre@inserm.fr (PL);
antonny.czarnecki@u-bordeaux.fr
(AC)

†These authors contributed
equally to this work
‡These authors also contributed
equally to this work

Competing interests: The
authors declare that no
competing interests exist.

Reviewing editor: Jeffrey C
Smith, National Institute of
Neurological Disorders and
Stroke, United States

**Abstract** Renshaw cells (V1$^R$) are excitable as soon as they reach their final location next to the spinal motoneurons and are functionally heterogeneous. Using multiple experimental approaches, in combination with biophysical modeling and dynamical systems theory, we analyzed, for the first time, the mechanisms underlying the electrophysiological properties of V1$^R$ during early embryonic development of the mouse spinal cord locomotor networks (E11.5–E16.5). We found that these interneurons are subdivided into several functional clusters from E11.5 and then display an unexpected transitory involution process during which they lose their ability to sustain tonic firing. We demonstrated that the essential factor controlling the diversity of the discharge pattern of embryonic V1$^R$ is the ratio of a persistent sodium conductance to a delayed rectifier potassium conductance. Taken together, our results reveal how a simple mechanism, based on the synergy of two voltage-dependent conductances that are ubiquitous in neurons, can produce functional diversity in embryonic V1$^R$ and control their early developmental trajectory.

## Introduction

The development of the central nervous system (CNS) follows complex steps, which depend on genetic and environmental factors and involve interactions between multiple elements of the neural tissue. Remarkably, emergent neurons begin to synchronize soon after the onset of synapse formation, generating long episodes of low-frequency (<0.01 Hz) correlated spontaneous network activity (SNA) (*O'Donovan, 1999*; *Saint-Amant, 2010*; *Blankenship and Feller, 2010*; *Myers et al., 2005*; *Milner and Landmesser, 1999*; *Hanson and Landmesser, 2003*; *Momose-Sato and Sato, 2013*; *Khazipov and Luhmann, 2006*). In the mouse embryonic spinal cord (SC), SNA is driven by an excitatory cholinergic-GABAergic loop between motoneurons (MNs) and interneurons (INs), GABA being

depolarizing before embryonic day 16.5 (E16.5) (*Allain et al., 2011*). SNA emerges around E12.5 (*Myers et al., 2005*; *Hanson and Landmesser, 2003*; *Branchereau et al., 2002*; *Yvert et al., 2004*; *Czarnecki et al., 2014*), at a time when functional neuromuscular junctions are not yet established (*Pun et al., 2002*), and sensory and supraspinal inputs have not yet reached the spinal motor networks (*Angelim et al., 2018*; *Marmigère and Ernfors, 2007*; *Ozaki and Snider, 1997*; *Ballion et al., 2002*).

Several studies pointed out that SNA is an essential component in neuronal networks formation (*Moody and Bosma, 2005*; *Spitzer, 2006*; *Katz and Shatz, 1996*; *Hanson et al., 2008*). In the SC, pharmacologically induced disturbances of SNA between E12.5 and E14.5 induce defects in the formation of motor pools, in motor axon guidance to their target muscles, and in the development of motor networks (*Myers et al., 2005*; *Hanson et al., 2008*; *Hanson and Landmesser, 2004*; *Hanson and Landmesser, 2006*). During SNA episodes, long-lasting giant depolarization potentials (GDPs) are evoked in the SC, mainly by the massive release of GABA onto MNs (*Czarnecki et al., 2014*). Immature Renshaw cells (V1$^R$) are likely the first GABAergic partners of MNs in the mouse embryo (*Benito-Gonzalez and Alvarez, 2012*; *Boeri et al., 2018*), and the massive release of GABA during SNA probably requires that many of them display repetitive action potential (AP) firing or plateau potential (PP) activity (*Boeri et al., 2018*).

However, little is known about the firing pattern of embryonic V1$^R$ and the maturation of their intrinsic properties. We recently found that V1$^R$ exhibit heterogeneous excitability properties when SNA emerges in the SC (*Boeri et al., 2018*) in contrast to adult Renshaw cells that constitute a functionally homogeneous population (*Perry et al., 2015*; *Bikoff et al., 2016*). Whether this early functional diversity really reflects distinct functional classes of V1$^R$, how this diversity evolves during development, and what are the underlying biophysical mechanisms remain open questions. The present study addresses these issues using multiple approaches, including patch-clamp recordings, cluster analysis, biophysical modeling, and dynamical systems theory. The firing patterns of V1$^R$ and the mechanisms underlying their functional diversity are analyzed during a developmental period covering the initial phase of development of SC activity in the mouse embryo (E11.5–E14.5), when SNA is present, and during the critical period (E14.5–E16.5), when GABAergic neurotransmission gradually shifts from excitation to inhibition (*Delpy et al., 2008*) and locomotor-like activity emerges (*Myers et al., 2005*; *Branchereau et al., 2002*; *Yvert et al., 2004*).

We discover that the balance between the slowly inactivating subthreshold persistent sodium inward current ($I_{Nap}$, *Crill, 1996*) and the delayed rectifier potassium outward current ($I_{Kdr}$), accounts for the heterogeneity of embryonic V1$^R$ and the changes in firing pattern during development. The heterogeneity of V1$^R$ at E12.5 arises from the existence of distinct functional groups. Surprisingly, and in opposition to the classically accepted development scheme (*Sillar et al., 1992*; *Gao and Ziskind-Conhaim, 1998*; *Gao and Lu, 2008*; *McKay and Turner, 2005*; *Liu et al., 2016*; *Pineda and Ribera, 2010*), we show that the embryonic V1$^R$ population loses its ability to support tonic firing from E13.5 to E15.5, exhibiting a transient functional involution during its development. Our experimental and theoretical results provide a global view of the developmental trajectories of embryonic V1$^R$. They demonstrate that a simple mechanism, based on the synergy of only two major opposing voltage-dependent currents, accounts for functional diversity in these immature neurons.

## Results

### The delayed rectifier potassium current I$_{Kdr}$ is a key partner of the persistent sodium current I$_{Nap}$ in controlling embryonic V1$^R$ firing patterns during development

We previously highlighted that V1$^R$ are spontaneously active at E12.5. Their response to a 2 s suprathreshold depolarizing current steps revealed four main patterns, depending on the recorded IN (*Boeri et al., 2018*): (1) single spiking (SS) V1$^R$ that fires only 1–3 APs at the onset of the depolarizing pulse, (2) repetitive spiking (RS) V1$^R$, (3) mixed events (ME) V1$^R$ that show an alternation of APs and PPs, or (4) V1$^R$ that displays a long-lasting sodium-dependent PP (*Figure 1A1–A4*).

We also uncovered a relationship between $I_{Nap}$ and the ability of embryonic V1$^R$ to sustain repetitive firing (*Boeri et al., 2018*). However, the heterogeneous firing patterns of V1$^R$ observed at E12.5 could not be fully explained by variations in $I_{Nap}$ (*Boeri et al., 2018*), suggesting the involvement of

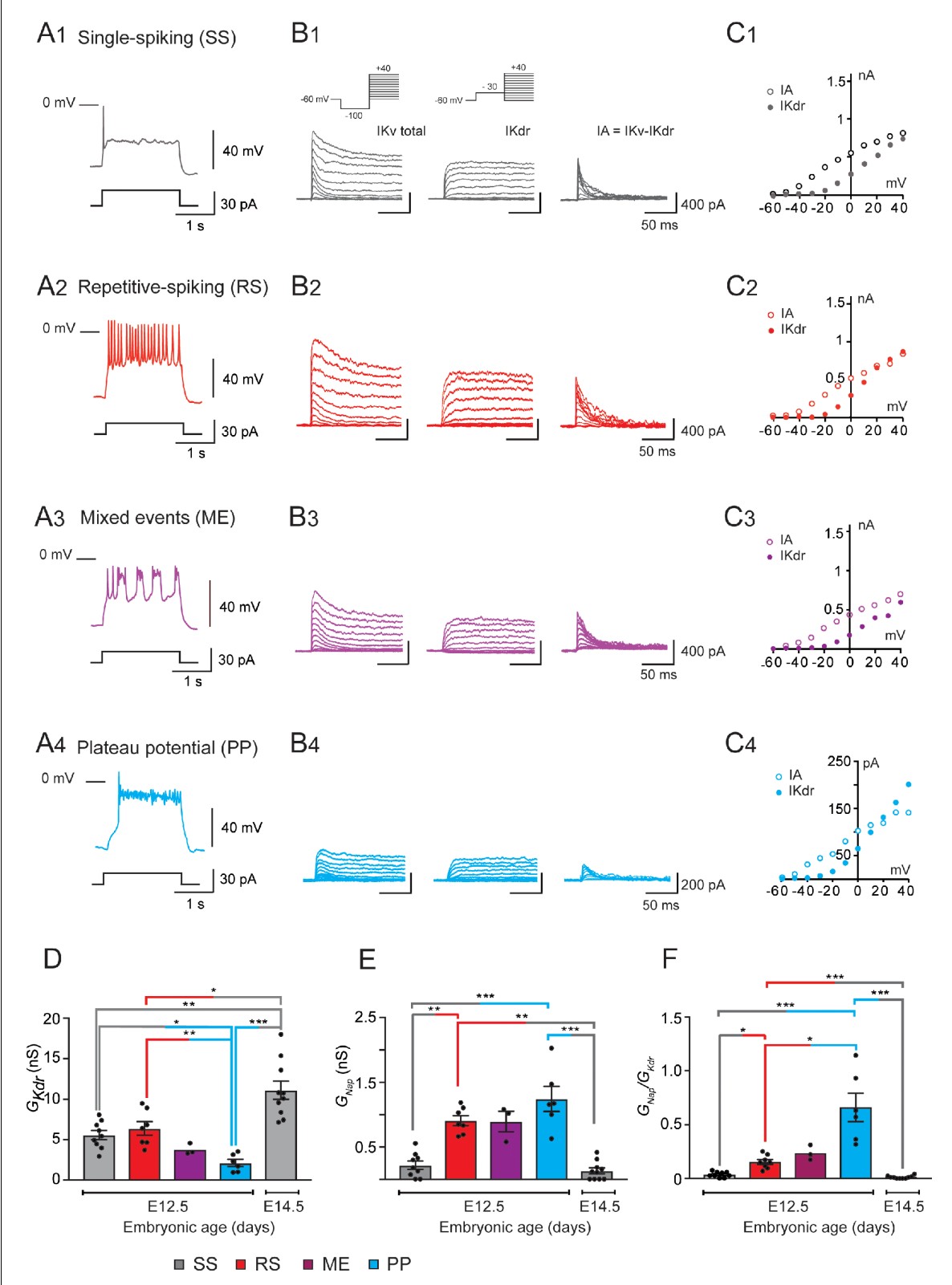

**Figure 1.** $G_{Kdr}$ and $G_{Nap}$ in embryonic V1[R] at E12.5 and E14.5. (**A**) Representative traces of voltage responses showing single spiking (SS) activity in E12.5 SS V1[R] (**A1**), repetitive action potential firing in repetitive spiking (RS) V1[R] (**A2**), mixed plateau potential activity (PP) and repetitive action potential firing in mixed event (ME) V1[R] (**A3**), and PP activity in PP V1[R] (**A4**). (**B**) Representative examples of the total outward K⁺ currents (IKV total) obtained from $V_H = -100$ mV (left traces), of $I_{Kdr}$ ($V_H = -30$ mV, middle traces), and of isolated $I_A$ (left traces) recorded at E12.5 in SS V1[R] (**B1**), RS V1[R] (**B2**), ME V1[R] (**B3**),

*Figure 1 continued on next page*

Figure 1 continued

and PP V1$^R$ (B4). Voltage-dependent potassium currents were evoked in response to 10 mV membrane potential steps (200 ms) from –100 or from –30 mV to +40 mV (10 s interval between pulses). V1$^R$ were voltage clamped at $V_H$ = –60 mV. A prepulse of –40 mV (300 ms) was applied to activate both $I_A$ and $I_{Kdr}$. $I_{Kdr}$ was isolated by applying a prepulse of -30 mV (300 ms) to inactivate $I_A$ (B1 inset). $I_A$ was isolated by subtracting step-by-step the currents obtained using a prepulse of -30 mV ($V_H$ = –30 mV) from the currents obtained using a prepulse of –40 mV ($V_H$ = –100 mV). (C) Current-voltage relationship ($I − V$ curves) of $I_{Kdr}$ (filled circles) and of $I_A$ (open circles) recorded in SS V1$^R$ (C1), RS V1$^R$ (C2), ME V1$^R$ (C3), and PP V1$^R$ (C4). $I − V$ curves were obtained from currents shown in (B1), (B2), (B3), and (B4). Note that $I − V$ curves are similar between SS V1$^R$, RS V1$^R$, ME V1$^R$, and PP V1$^R$. (D) Bar graph showing maximal $G_{Kdr}$ value (Max $G_{Kdr}$) in SS V1$^R$ at E12.5 (n = 9; N = 9; gray bar) and at E14.5 (n = 10; N = 10 gray bar), and in RS V1$^R$ (n = 7; N = 7; red bar), ME V1$^R$ (n = 3; N = 3 purple bar), and PP V1$^R$ at E12.5 (n = 7; N = 7 blue bar) was calculated from $I_{Kdr}$ at $V_H$ = + 20 mV, assuming a K$^+$ equilibrium potential of –96 mV. There is no significant difference in $G_{Kdr}$ between SS V1$^R$ and RS V1$^R$, while $G_{Kdr}$ is significantly smaller in PP V1$^R$ as compared to $G_{Kdr}$ SS V1$^R$ and RS V1$^R$ was significantly higher in SS V1$^R$ at E14.5 than in SS V1$^R$, RS V1$^R$, and PP V1$^R$ at E12.5 (Kruskal–Wallis test p<0.0001; SS V1$^R$ versus RS V1$^R$ at E12.5, p=0.5864; SS V1$^R$ versus PP V1$^R$ at E12.5, p=0.0243; RS V1$^R$ versus PP V1$^R$ at E12.5, p=0.0086; E14.5 SS V1$^R$ versus E12.5 SS V1$^R$, p=0.0048; E14.5 SS V1$^R$ versus E12.5 RS V1$^R$, p=0.0384, E14.5 SS V1$^R$ versus E12.5 PP V1$^R$, p<0.0001). The increase in $G_{Kdr}$ density between E12.5 and E14.5 is likely to be due to the increase in neuronal size (input capacitance; *Figure 2A*). Indeed, there was no significant difference (Mann–Whitney test, p=0.133) in $G_{Nap}$ between SS V1$^R$ at E12.5 (n = 9; N = 9 gray bar) and at E14.5 (n = 10; N = 10 gray bar). (E) Bar graph showing the maximal Max $G_{Nap}$ value ($G_{Nap}$) in SS V1$^R$ at E12.5 (n = 9; N = 9 gray bar) and E14.5 (n = 10; N = 10 gray bar), and in RS V1$^R$ (n = 8; N = 8 red bar), ME V1$^R$ (n = 3; N = 3 purple bar), and PP V1$^R$ (n = 6; N = 6 blue bar) at E12.5. Max $I_{Nap}$ was calculated from maximal $G_{Nap}$ value measured on current evoked by assuming a Na$^+$ equilibrium potential of +60 mV. There was no difference in $G_{Nap}$ between RS V1$^R$ and PP V1$^R$. On the contrary, $G_{Nap}$ measured in SS V1$^R$ at E12.5 or at E14.5 was significantly smaller as compared to $G_{Nap}$ measured at E12.5 in RS V1$^R$ or in PP V1$^R$ measured at E12.5 and E14.5 in SS V1$^R$ were not significantly different (Kruskal–Wallis test p<0.0001; E12.5 SS V1$^R$ versus E12.5 RS V1$^R$, p=0.0034; E12.5 SS V1$^R$ versus E12.5 PP V1$^R$, p=0.0006; E12.5 RS V1$^R$ versus E12.5 PP V1$^R$, p=0.5494; E14.5 SS V1$^R$ versus E12.5 SS V1$^R$, p=0.5896; E14.5 SS V1$^R$ versus E12.5 RS V1$^R$, p=0.0005; E14.5 SS V1$^R$ versus E12.5 PP V1$^R$, p<0.0001). (F) Histograms showing the $G_{Kdr}$ / $G_{Nap}$ ratio in SS V1$^R$ at E12.5 (n = 9; gray bar) and E14.5 (n = 10; green bar) and in RS V1$^R$ (n = 8; red bar), ME V1$^R$ (n = 3; purple bar), and PP V1$^R$ (n = 6; blue bar) at E12.5. Note that the $G_{Kdr}$ / $G_{Nap}$ ratio differs significantly between SS V1$^R$, RS V1$^R$, and PP V1$^R$ at E12.5, while it is not different between SS V1$^R$ recorded at E12.5 and at E14.5 (Kruskal–Wallis test p<0.0001; SS V1$^R$ versus RS V1$^R$ at E12.5, p=0.0367; SS V1$^R$ versus PP V1$^R$ at E12.5, p<0.0001; RS V1$^R$ versus PP V1$^R$ at E12.5, p=0.0159; E14.5 SS V1$^R$ versus E12.5 SS V1$^R$, p=0.2319; E14.5 SS V1$^R$ versus E12.5 RS V1$^R$, p=0.0017; E14.5 SS V1$^R$ versus E12.5 PP V1$^R$p<0.0001). Data shown in (A) and (B) were used to calculate $G_{Kdr}/C_{in}$ ratio shown in (C) (*p<0.05, **p<0.01, ***p<0.001).

other voltage-gated channels in the control of the firing pattern of V1$^R$, in particular potassium channels, known to control firing and AP repolarization. Our voltage clamp protocol, performed in the presence of tetrodotoxin (TTX) (1 µM), did not disclose any inward rectifying current (hyperpolarizing voltage steps to –100 mV from $V_H$ = –20 mV, data not shown), but revealed two voltage-dependent outward potassium currents, a delayed rectifier current ($I_{Kdr}$), and a transient potassium current ($I_A$) in all embryonic V1$^R$, whatever the firing pattern; *Figure 1B1–B4*. These currents are known to control AP duration ($I_{Kdr}$) or firing rate ($I_A$), respectively (*Coetzee et al., 1999*). The activation threshold of $I_{Kdr}$ lied between –30 mV and –20 mV and the threshold of $I_A$ between –60 mV and –50 mV, (n = 27; N = 27 embryos) (*Figure 1C1–C4*). Removing external calcium had no effect on potassium current I/V curves (data not shown), suggesting that calcium-dependent potassium currents are not yet present at E12.5.

It was unlikely that the heterogeneity of V1$^R$ firing patterns resulted from variations in the intensity of $I_A$. Indeed, its voltage-dependent inactivation (time constant: 23.3 ± 2.6 ms, n = 8; N = 8), which occurs during the depolarizing phase of an AP, makes it ineffective to control AP or PP durations. This was confirmed by our theoretical analysis (*Figure 7—figure supplement 1*). We thus focused our study on $I_{Kdr}$. At E12.5, PP V1$^R$ had a significantly lower $G_{Kdr}$ (2.12 ± 0.44 nS, n = 6; N = 6) than SS V1$^R$ (5.57 ± 0.56 nS, n = 9; N = 9) and RS V1$^R$ (6.39 ± 0.83 nS, n = 7; N = 7) (*Figure 1D*). However, there was no significant difference in $G_{Kdr}$ between SS V1$^R$ and RS V1$^R$ at E12.5 (*Figure 1D*), which indicated that variations in $G_{Kdr}$ alone could not explain all the firing patterns observed at E12.5. Similarly, there was no significant difference in $G_{Nap}$ between RS V1$^R$ (0.91 ± 0.21nS, n = 8; N = 8) and PP V1$^R$ (1.24 ± 0.19 nS, n = 6; N = 6) at E12.5 (*Figure 1E*), indicating that variations in $G_{Nap}$ alone could not explain all the firing patterns of V1$^R$ at E12.5 (*Boeri et al., 2018*). In contrast, $G_{Nap}$ measured in SS V1$^R$ at E12.5 (0.21 ± 0.20 nS, n = 9; N = 9) were significantly lower compared to $G_{Nap}$ measured in RS V1$^R$ and in PP V1$^R$ at E12.5 (*Figure 1E*).

Mature neurons often display multiple stable firing patterns (*O'Leary et al., 2013*; *Taylor et al., 2009*; *Alonso and Marder, 2019*). This usually depends on the combination of several outward and inward voltage- or calcium-dependent conductances and on their spatial localization (*O'Leary et al., 2013*; *Taylor et al., 2009*; *Alonso and Marder, 2019*). In contrast, immature V1$^R$ have a limited

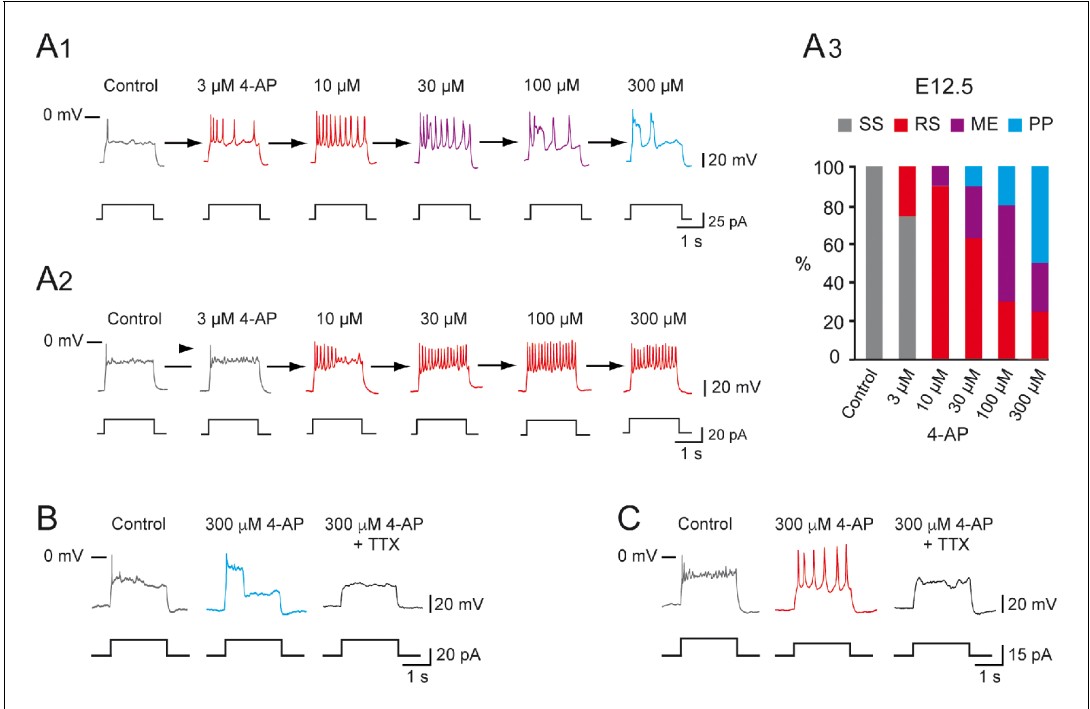

**Figure 2.** Increasing 4-aminopiridine (4-AP) concentration changed the firing pattern of single spiking (SS) embryonic V1$^R$ recorded at E12.5. The firing pattern of embryonic V1$^R$ was evoked by 2 s suprathreshold depolarizing current steps. (**A**) Representative traces showing examples of the effect of increasing concentration of 4-AP (from 3 to 300 μM) on the firing pattern of a SS V1$^R$ recorded at E12.5. Note that in (**A1**) increasing 4-AP concentration converted SS (gray trace) to repetitive spiking (red trace), repetitive spiking to a mixed event pattern (purple trace), and mixed events to plateau potential (blue trace). (**A2**) Example of SS V1$^R$ in which increasing 4-AP concentration converted SS to repetitive spiking only. (**A3**) Bar plots showing the change in the firing pattern of SS V1$^R$ according to 4-AP concentrations (control n = 10; N = 10, 3 μM 4-AP n = 8; N = 8, 10 μM 4-AP n = 10; N = 10, 30 μM 4-AP n = 10; N = 10, 100 μM 4-AP n = 10; N = 10, 300 μM 4-AP n = 8; N = 8). (**B**) Representative traces showing the effect of 0.5 μM tetrodotoxin (TTX) on a plateau potential evoked in a SS V1$^R$ in the presence of 300 μM 4-AP. (**C**) Representative traces showing the effect of 0.5 μM TTX on repetitive action potential firing evoked in a SS V1$^R$ in the presence of 300 μM 4-AP. In both cases, the application of TTX fully blocked the responses evoked in the presence of 4-AP, indicating that they were underlain by the activation of voltage-gated Na$^+$ channels.

The online version of this article includes the following figure supplement(s) for figure 2:

**Figure supplement 1.** Effect of 4-aminopiridine (4-AP) on $I_{Kdr}$ and $I_A$ in embryonic V1$^R$.

**Figure supplement 2.** Effect of 4-aminopiridine (4-AP) application in repetitively spiking (RS).

repertoire of voltage-dependent currents ($I_{Nat}$ and $I_{Nap}$, $I_{Kdr}$ and $I_A$) at E12.5, and we did not find any evidence of voltage-dependent calcium currents at this age (*Boeri et al., 2018*). Blocking $I_{Nap}$ prevented PP activity, PP-V1$^R$ becoming unexcitable, and turned RS V1$^R$ into SS V1$^R$ (*Boeri et al., 2018*). Therefore, we hypothesized that the different firing patterns of V1$^R$ observed at E12.5 were related to the $G_{Nap}/G_{Kdr}$ ratio only, with variations in the intensity of $I_A$ being unlikely to account for the heterogeneity of firing pattern. We found that this ratio was significantly lower for SS V1$^R$ recorded at E12.5 ($G_{Nap}/G_{Kdr}$ = 0.043 ± 0.015, n = 9) compared to RS V1$^R$ (0.154 ± 0.022, n = 8) and PP V1$^R$ (0.66 ± 0.132, n = 6) (*Figure 1F*). We also found that the $G_{Nap}/G_{Kdr}$ ratio was significantly lower for RS V1$^R$ compared to PP V1$^R$ (*Figure 1F*).

Altogether, these results strongly suggest that, although the presence of $I_{Nap}$ is required for embryonic V1$^R$ to fire repetitively or to generate PPs (*Boeri et al., 2018*), the heterogeneity of the firing pattern observed between E12.5 is not determined by $I_{Nap}$ per se but likely by the balance between $I_{Nap}$ and $I_{Kdr}$.

## Manipulating the balance between G$_{Nap}$ and G$_{Kdr}$ changes embryonic V1$^R$ firing patterns

We previously showed that blocking $I_{Nap}$ with riluzole converted PP V1$^R$ or RS V1$^R$ into SS V1$^R$ (*Boeri et al., 2018*). To confirm further that the balance between $G_{Nap}$ and $G_{Kdr}$ was the key factor in

the heterogeneity of V1$^R$ firing patterns, we assessed to what extent a given E12.5 SS V1$^R$ cell could change its firing pattern when $I_{Kdr}$ was gradually blocked by 4-aminopiridine (4-AP). We found that $I_{Kdr}$ could be blocked by micromolar concentrations of 4-AP without affecting $I_A$ (*Figure 2—figure supplement 1*). 4-AP, applied at concentrations ranging from 0.3 µM to 300 µM, specifically inhibited $I_{Kdr}$ with an IC$_{50}$ of 2.9 µM (*Figure 2—figure supplement 1C1*).

We then determined to what extent increasing the concentration of 4-AP modified the firing pattern of V1$^R$ at E12.5. Applying 4-AP at concentrations ranging from 3 µM to 300 µM changed the firing pattern of SS V1$^R$ (n = 10; N = 10) in a concentration-dependent manner (*Figure 2A1–A3*). In 50% of the recorded V1$^R$, increasing 4-AP concentrations successfully transformed SS V1$^R$ into PP V1$^R$ with the following sequence: SS → RS → ME → PP (*Figure 2A1*). In a second group of embryonic V1$^R$ (25%), 4-AP application only evoked mixed activity, with the same sequence as aforementioned (SS → RS → ME) (data not shown). In the remaining SS V1$^R$ (25%), increasing 4-AP concentration only led to sustained AP firing (*Figure 2A2*). Application of 300 µM 4-AP on RS V1$^R$ at E12.5 evoked MEs or PPs (*Figure 2—figure supplement 2*). PPs and RS evoked in the presence of 300 µM 4-AP were fully blocked by 0.5–1 µM TTX, indicating that they were generated by voltage-gated Na$^+$ channels (*Figure 2B, C, Figure 2—figure supplement 2*). It should be noted that the application of 300 µM of 4-AP induced a significant 30.5 ± 12.4% increase (p=0.0137; Wilcoxon test) of the input resistance (1.11 ± 0.08 GΩ versus 1.41 ± 0.12 GΩ; n = 11; N = 11).

These results show that, in addition to $I_{Nap}$, $I_{Kdr}$ is also a major determinant of the firing pattern of embryonic V1$^R$. The above suggests that the firing patterns depend on a synergy between $I_{Nap}$ and $I_{Kdr}$ and that the different patterns can be ordered along the following sequence SS → RS → ME → PP when the ratio $G_{Nap}/G_{Kdr}$ is increased.

## The heterogeneity of the V1$^R$ firing patterns decreases during embryonic development

It was initially unclear whether these different firing patterns corresponded to well-separated classes within the E12.5 V1$^R$ population or not. To address this question, we performed a hierarchical cluster analysis on 163 embryonic V1$^R$ based on three quantitative parameters describing the firing pattern elicited by the depolarizing pulse: the mean duration of evoked APs or PPs measured at half-amplitude (mean ½Ad), the variability of the event duration during repetitive firing (coefficient of variation of ½Ad [CV ½Ad]), and the total duration of all events, expressed in percentage of the pulse duration (depolarizing duration ratio [ddr]) (*Figure 3A* insets). In view of the large dispersion of mean ½Ad and ddr values, cluster analysis was performed using the (decimal) logarithm of these two quantities (*Sigworth and Sine, 1987*). The analysis of the distribution of log mean ½Ad, CV ½Ad, and log ddr revealed multimodal histograms that could be fitted with several Gaussians (*Figure 3—figure supplement 1A1–C1*). Cluster analysis based on these three parameters showed that the most likely number of clusters was 5 (*Figure 3A, B*), as determined by the silhouette width measurement (*Figure 3B*). Two clearly separated embryonic V1$^R$ groups with CV ½Ad = 0 stood out, as shown in the 3D plot in *Figure 3C*. The cluster with the largest ½Ad (mean ½Ad = 833.5 ± 89.99 ms) and the largest ddr (0.441 ± 0.044) contained all PP V1$^R$ (n = 35; N = 29) (*Figure 3C, D, Figure 3—figure supplement 1A2, C2*). Similarly, the cluster with the shortest ½Ad (9.73 ± 0.66 ms) and the lowest ddr (0.0051 ± 0.0004) contained all SS V1$^R$ (n = 46; N = 37) (*Figure 3C, D, Figure 3—figure supplement 1A2, C2*).

The three other clusters corresponded to V1$^R$ with nonzero values of CV ½Ad (*Figure 3C*). A first cluster regrouping all RS V1$^R$ (n = 69; N = 61) was characterized by smaller values of ½Ad (23.91 ± 1.43 ms), CV ½Ad (27.36 ± 1.64%), and ddr (0.11 ± 0.01) (*Figure 3C–D, Figure 3—figure supplement 1A2, C2*). The last two clusters corresponded to ME V1$^R$ (*Figure 3C, D*). The smaller cluster, characterized by a larger CV ½Ad (170.9 ± 8.9%; n = 4; N = 4), displayed a mix of APs and short PPs, while the second cluster, with smaller CV ½Ad (87.61 ± 7.37%; n = 9; N = 9), displayed a mix of APs and long-lasting PPs (*Figure 3D, Figure 3—figure supplement 1B2*). Their ½Ad and ddr values were not significantly different (*Figure 3—figure supplement 1A2, C2*).

It must be noted that three embryonic V1$^R$ (1.8%) were apparently misclassified since they were aggregated within the RS cluster although having zero CV ½Ad (*Figure 3C*, arrows). Examination of their firing pattern revealed that this was because they generated only two APs, although their ddr (0.16–0.2) and ½ Ad values (31.6–40.3 ms) were well in the range corresponding tto the RS cluster.

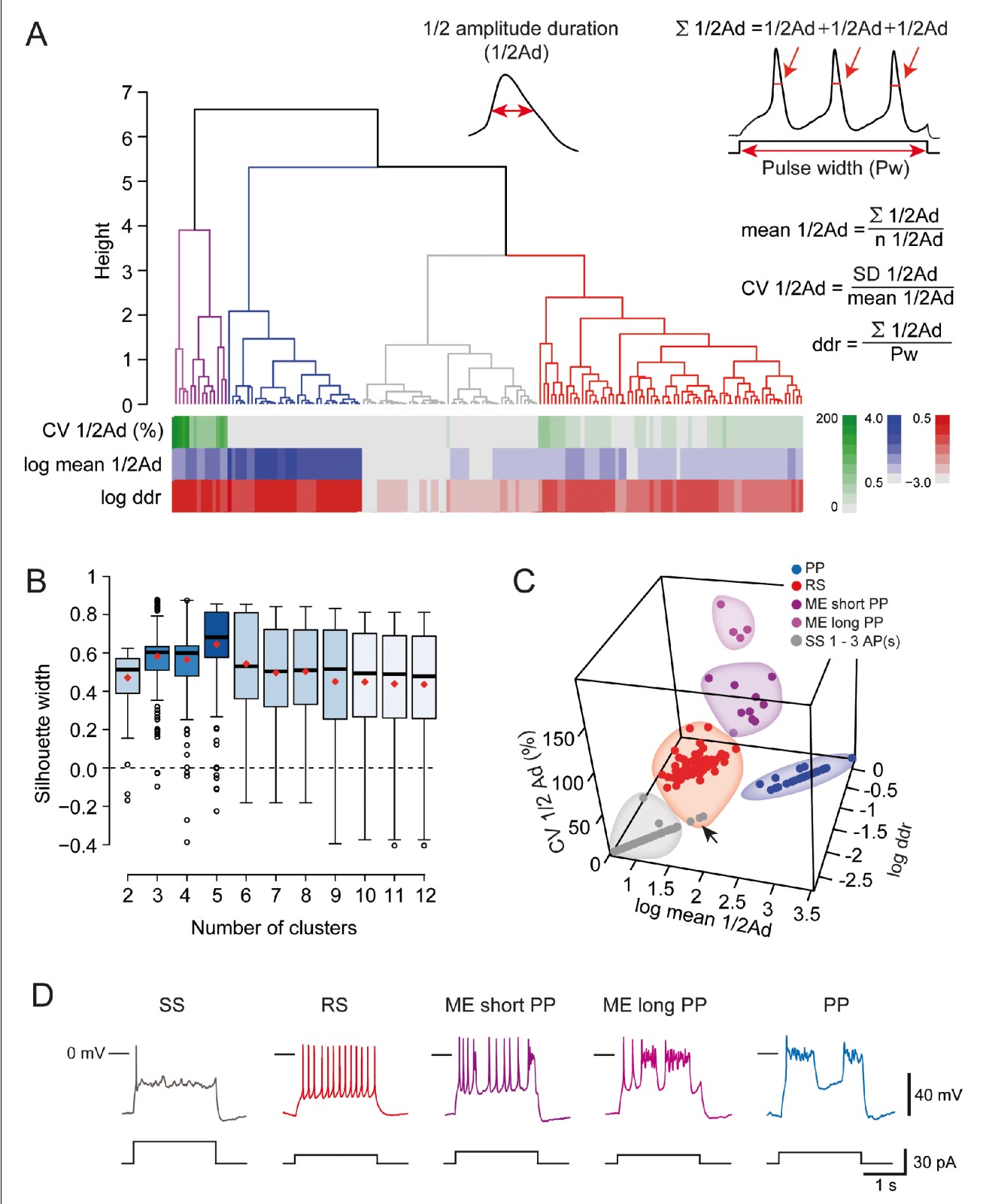

**Figure 3.** Cluster analysis of V1$^R$ firing pattern at E12.5. (A, insets) Cluster analysis of embryonic V1$^R$ firing pattern was performed using three parameters that describe the firing pattern during a 2 s suprathreshold depolarizing pulses: the mean of the half-amplitude event duration (mean ½Ad), the coefficient of variation of ½ Ad (CV ½Ad) allowing to quantify the action potential variation within a train (CV was set to 0 when the number of spikes evoked by a depolarizing pulse was ≤3) and the duration ratio ddr = Σ½ Ad/Pw, obtained by dividing the sum of ½ Ad by the pulse duration

*Figure 3 continued on next page*

*Figure 3 continued*

Pw, that indicates the total time spent in the depolarized state. For example, ddr = 1 when a plateau potential (PP) lasts as long as the depolarizing pulse. Conversely, its value is low when the depolarizing pulse evokes a single AP only. (**A**) Dendrogram for complete linkage hierarchical clustering of 164 embryonic V1$^R$ (N = 140) according to the values of log mean ½Ad, CV ½Ad, and log ddr. The colored matrix below the dendrogram shows the variations of these three parameters for all the cells in the clusters (colored trees) extracted from the dendrogram. (**B**) The number of clusters was determined by analyzing the distribution of silhouette width values (see Materials and methods). The box plots show the distribution of silhouette width values when the number of clusters k varies from 2 to 12. The mean silhouette width values (red diamond-shaped points) attained their maximum when the estimated cluster number was 5. (**C**) 3D plot showing cluster distribution of embryonic V1$^R$ according to log mean ½Ad, CV ½Ad, and log ddr. Each cluster corresponds to a particular firing pattern as illustrated in (**D**). V1$^R$ that cannot sustain repetitive firing of APs (1–3 AP/pulse only, gray, single spiking [SS]), V1$^R$ that can fire tonically (red, repetitive spiking [RS]), V1$^R$ with a firing pattern characterized by a mix of APs and relatively short PPs (dark purple, mixed event short PP [ME short PP]), V1$^R$ with a firing pattern characterized by a mix of APs and relatively long plateau potentials (light purple, mixed event long PP [ME long PP]), and V1$^R$ with evoked PPs only (blue, PP). The arrow in (**C**) indicates three misclassified V1$^R$ that could not sustain repetitive firing although they were assigned to the cluster of repetitively firing V1$^R$ (see text).

The online version of this article includes the following source data and figure supplement(s) for figure 3:

**Source data 1.** Numerical data used to perform cluster analysis shown in *Figure 3*.
**Figure supplement 1.** Distributions of log half-amplitude event duration (log ½Ad), coefficient of variation of ½ Ad (CV ½Ad), and log depolarizing duration ratio (log ddr) values related to the cluster analysis of embryonic V1$^R$ firing patterns.

These different firing patterns of V1$^R$ might reflect different states of neuronal development (*Gao and Ziskind-Conhaim, 1998*; *Ramoa and McCormick, 1994*; *Belleau and Warren, 2000*; *Picken Bahrey and Moody, 2003*). SS and/or PPs are generally believed to be the most immature forms of firing pattern, RS constituting the most mature form (*Spitzer, 2006*; *Tong and McDearmid, 2012*). If it were so, the firing patterns of embryonic V1$^R$ would evolve during embryonic development from SS or PP to RS, this latter firing pattern becoming the only one in neonates (*Perry et al., 2015*) and at early postnatal stages (*Bikoff et al., 2016*). However, RS neurons already represent 41% of V1$^R$ at E12.5. We therefore analyzed the development of firing patterns from E11.5, when V1$^R$ terminate their migration and reach their final position (*Alvarez et al., 2013*), to E16.5. This developmental period covers a first phase of development (E11.5–E14.5), where lumbar spinal networks exhibit SNA, and a second phase (E14.5–E16.5), where locomotor-like activity emerges (*Myers et al., 2005*; *Yvert et al., 2004*; *Allain et al., 2010*; *Branchereau et al., 2000*). We first analyzed changes in the intrinsic properties (input capacitance $C_{in}$, input resistance $R_{in} = 1/G_{in}$, and spike voltage threshold) of V1$^R$. $C_{in}$ did not change significantly from E11.5 to E13.5 (*Figure 4A1*), remaining of the order of 12 pF, in agreement with our previous work (*Boeri et al., 2018*). However, it increased significantly at the transition between the two developmental periods (E13.5–E15.5) to reach about 23.5 pF at E15.5 (*Figure 4A1*). A similar developmental pattern was observed for $R_{in}$, which remained stable during the first phase from E11.5 to E14.5 ($R_{in} ≈ $ 1–1.2 GΩ) but decreased significantly after E14.5 to reach about 0.7 GΩ at E15.5 (*Figure 4A2*). Spike threshold also decreased significantly between the first and the second developmental phases, dropping from about –34 mV at E12.5 to about –41 mV at E16.5 (*Figure 4A3*). Interestingly, this developmental transition around E14.5 corresponds to the critical stage at which SNA gives way to a locomotor-like activity (*Yvert et al., 2004*; *Allain et al., 2010*; *Branchereau et al., 2000*) and rhythmic activity becomes dominated by glutamate release rather than acetylcholine release (*Myers et al., 2005*).

This led us to hypothesize that this developmental transition could be also critical for the maturation of V1$^R$ firing patterns. The distinct firing patterns observed at E12.5 were already present at E11.5 (*Figure 4B1, C*), but the percentage of RS V1$^R$ strongly increased from E11.5 to E12.5, while the percentage of ME V1$^R$ decreased significantly (*Figure 4C*). The heterogeneity of V1$^R$ firing patterns then substantially diminished. PPs were no longer observed at E13.5 (*Figure 4B2, C*) and ME V1$^R$ disappeared at E14.5 (*Figure 4B3, C*). Interestingly, the proportion of SS V1$^R$ remained high from E13.5 to E15.5 and even slightly increased (91.23% at E14.5% and 93.33% at E15.5; *Figure 4C*). This trend was partially reversed at E16.5 as the percentage of RS V1$^R$ increased at the expense of SS V1$^R$ (67.86% SS V1$^R$ and 32.34% RS V1$^R$; *Figure 4B5, C*). This decrease in repetitive firing capability after E13.5 was surprising in view of what is classically admitted on the developmental pattern of neuronal excitability (*Moody and Bosma, 2005*; *Spitzer et al., 2000*). Therefore, we verified that it did not reflect the death of some V1$^R$ after E13.5. Our data did not reveal any activated caspase3 (aCaspase3) staining in V1$^R$ (FoxD3 staining) at E14.5 (n = 10 SCs; N = 10) (*Figure 5*),

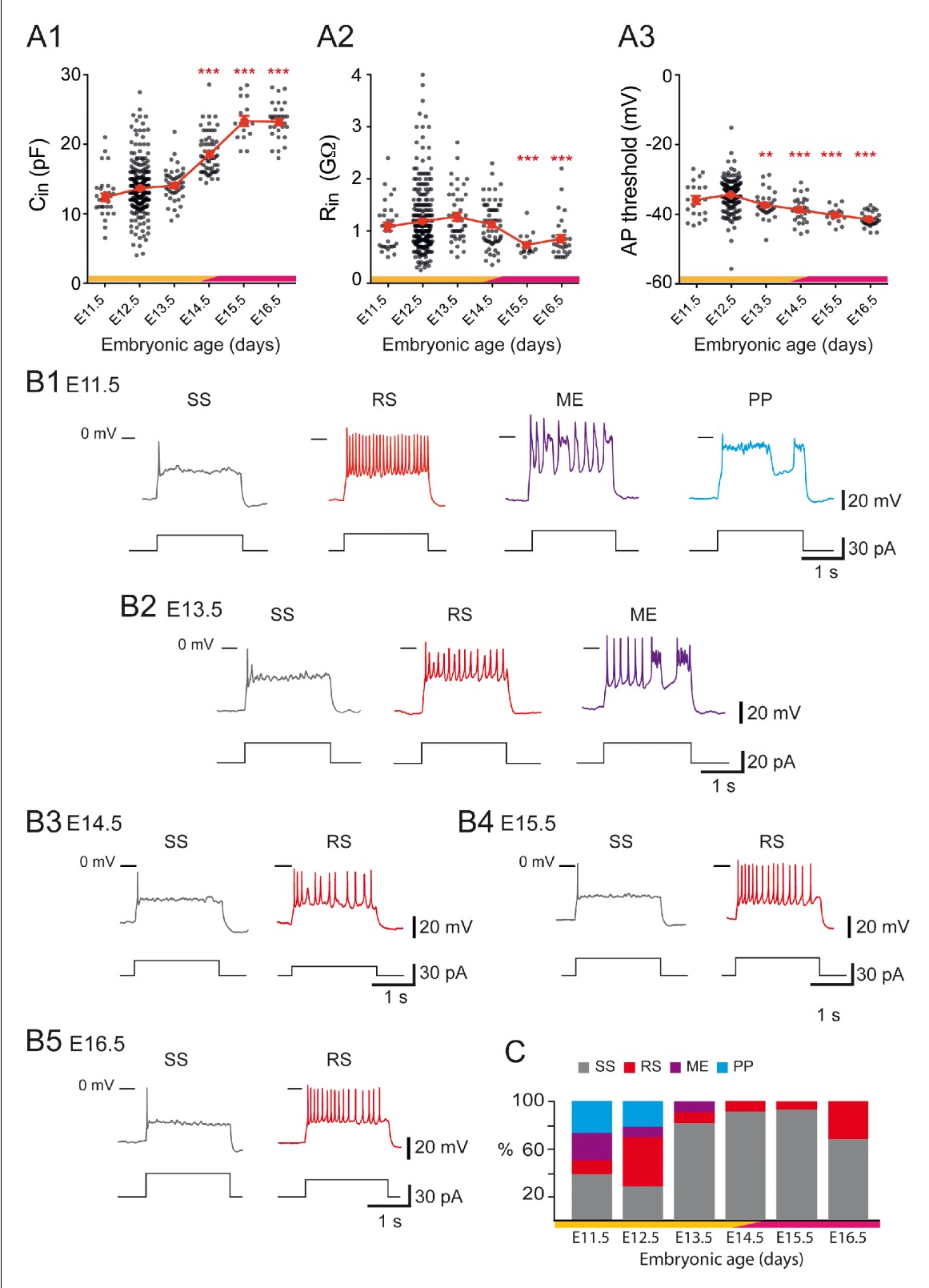

**Figure 4.** Developmental changes of embryonic V1$^R$ firing patterns from E11.5 to E16.5. (A1) Graph showing how the input capacitance $C_{in}$ of V1$^R$ changes with embryonic age. $C_{in}$ significantly increased between E12.5 or E13.5 and E14.5 (Kruskal–Wallis test p<0.0001; E12.5 versus E11.5 p=0.258, E12.5 versus E13.5 p=0.904, E12.5 versus E14.5 p<0.0001, E12.5 versus E15.5 p<0.0001, E12.5 versus E16.5 p<0.0001, E13.5 versus E14.5 p<0.0001, E13.5 versus E15.5 p<0.0001, E13.5 versus E16.5 p<0.0001; E11.5 n = 31; N = 27, E12.5 n = 267; N = 152, E13.5 n = 43; N = 40, E14.5 n = 61; N = 49, E15.5 n

*Figure 4 continued on next page*

*Figure 4 continued*

= 16; N = 4, E16.5 n = 30; N = 9). (**A2**) Graph showing how the input resistance $R_{in}$ of V1$^R$ changes with embryonic age. $R_{in}$ significantly decreased between E12.5 or E14.5 and E15.5 (Kruskal–Wallis test p<0.0001; E12.5 versus E11.5 p>0.999, E12.5 versus E13.5 p=0.724, E12.5 versus E14.5 p>0.999, E12.5 versus E15.5 p=0.0004, E12.5 versus E16.5 p=0.0005, E14.5 versus E15.5 p=0.0019, E14.5 versus E16.5 p<0.0058; E11.5 n = 31, E12.5 n = 261; N = 146, E13.5 n = 43; N = 40, E14.5 n = 60; N = 48, E15.5 n = 16; N = 4, E16.5 n = 30; N = 9). (**A3**) Graph showing how the threshold of regenerative events (action potentials [APs] and plateau potentials [PP]) of V1$^R$ changes with embryonic age. The average threshold became significantly more hyperpolarized after E12.5 (Kruskal–Wallis test p<0.0001; E12.5 versus E11.5 p=0.676, E12.5 versus E13.5 p=0.0039, E12.5 versus E14.5 p<0.0001, E12.5 versus E15.5 p<0.0001, E12.5 versus E16.5 p<0.0001, E13.5 versus E14.5 p>0.999, E13.5 versus E15.5 p=0.1398, E13.5 versus E16.5 p=0.0013; E14.5 versus E15.5 p>0.999, E14.5 versus E16.5 p=0.0634, E15.5 versus E16.5 p>0.999; E11.5 n = 20; N = 16, E12.5 n = 162; N = 139, E13.5 n = 31; N = 28, E14.5 n = 30; N = 26, E15.5 n = 16; N = 4, E16.5 n = 30; N = 9). Yellow and purple bars below the graphs indicate the two important phases of the functional development of spinal cord networks. The first one is characterized by synchronized neuronal activity (SNA), and the second one is characterized by the emergence of a locomotor-like activity (see text). Note that changes in $C_{in}$ and $R_{in}$ occurred at the end of the first developmental phase (*p<0.05, **p<0.01, ***p<0.001; control, E12.5). The intrinsic activation properties were analyzed using 2 s suprathreshold depolarizing current steps. (**B**) Representative traces of voltage responses showing single spiking (SS) V1$^R$ (gray), repetitive spiking (RS) V1$^R$ (red), mixed events (ME) V1$^R$ (purple), and PP V1$^R$ (blue) at E11.5 (**B1**), E13.5 (**B2**), E14.5 (**B3**) E15.5 (**B4**), and E16.5 (**B5**). (**C**) Bar graph showing how the proportions of the different firing patterns change from E11.5 to E16.5 (E11.5 n = 22; N = 18, E12.5 n = 163; N = 140, E13.5 n = 32; N = 29, E14.5 n = 57; N = 45, E15.5 n = 15; N = 4, E16.5 n = 28; N = 9). Yellow and purple bars below the graphs indicate the first and the second phase of functional embryonic spinal cord networks. The proportions of the different firing patterns significantly changed between E11.5 to E12.5 (Fisher's exact test, p=0.0052) with a significant increase in the proportion of RS V1$^R$ (Fisher's exact test, p=0.0336) and a significant decrease in the proportion of ME V1$^R$ (Fisher's exact test, p=0.01071) at E12.5. Only two firing patterns (SS and RS) were observed after E13.5 and most embryonic V1$^R$ lost their ability to sustain tonic firing after E13.5. However, at E16.5 the proportion of RS V1$^R$ significantly increased at the expense of SS V1$^R$ when compared to E14.5 (Fisher's exact test, p=0.0112), indicating that embryonic V1$^R$ began to recover the ability to sustain tonic firing after E15.5.

in agreement with previous reports showing that developmental cell death of V1$^R$ does not occur before birth (*Prasad et al., 2008*).

To determine whether $G_{Nap}$ and $G_{Kdr}$ also controlled the firing pattern of V1$^R$ at E14.5 (see *Figure 4B3, C*), we assessed the presence of $I_{Nap}$ and $I_{Kdr}$ in SS V1$^R$ at this embryonic age. Both $I_{Nap}$ and $I_{Kdr}$ were present in V1$^R$ at E14.5 (*Figure 6—figure supplement 1*, *Figure 6—figure supplement 2*), whereas, as in V1$^R$ at E12.5, no calcium-dependent potassium current was detected at this developmental age (not shown). In SS V1$^R$, $G_{Kdr}$ was significantly higher at E14.5 (11.11 ± 1.12 nS, n = 10; N = 10) than at E12.5 (*Figure 1D*). In contrast, $G_{Nap}$ was similar at E14.5 (0.13 ± 0.14 nS, n = 10; N = 10) and E12.5 (*Figure 1E*). We also found that the $G_{Nap}/G_{Kdr}$ ratio was significantly lower for SS V1$^R$ recorded at E14.5 (0.012 ± 0.004, n = 10) compared to RS V1$^R$ (0.154 ± 0.022, n = 8) and PP V1$^R$ (0.66 ± 0.132, n = 6) recorded at E12.5 (*Figure 1F*).

We tested the effect of 4-AP in SS V1$^R$ at E14.5. At this embryonic age, 300 µM 4-AP inhibited only 59.2% of $I_{Kdr}$. Increasing 4-AP concentration to 600 µM did not inhibit $I_{Kdr}$ significantly more (60.2%) (*Figure 6—figure supplement 2*), indicating that inhibition of $I_{Kdr}$ by 4-AP reached a plateau at around 300 µM. 600 µM 4-AP application had no significant effect on $I_A$ (*Figure 6—figure supplement 2*). The application of the maximal concentration of 4-AP tested (600 µM) converted SS V1$^R$ (n = 13; N = 13) to PP V1$^R$ (23.1%; *Figure 6A1, B*), RS V1$^R$ (38.5%; *Figure 6A2, B*), or ME V1$^R$ (38.4%; *Figure 6B*), as was observed at E12.5, thus indicating that the firing pattern of V1$^R$ depends on the balance between $I_{Nap}$ and $I_{Kdr}$ also at E14.5. PP and RS recorded in the presence of 4-AP at E14.5 were fully blocked by 0.5–1 µM TTX, indicating that they were generated by voltage-gated sodium channels (*Figure 6A1, A2*), as observed at E12.5.

## Theoretical analysis: the basic model

As shown in *Figure 7A* for 26 cells, in which both $G_{Nap}$ and $G_{Kdr}$ were measured, the three largest clusters revealed by the hierarchical clustering analysis (SS, RS, and PP, which account together for the discharge of more than 95% of cells, see *Figure 3*) correspond to well-defined regions of the $G_{Nap}$ - $G_{Kdr}$ plane. SS is observed only when $G_{Nap}$ is smaller than 0.6 nS. For larger values of $G_{Nap}$, RS occurs when $G_{Kdr}$ is larger than 3.5 nS, and V1$^R$ display PPs when $G_{Kdr}$ is smaller than 3.5 nS. ME (4.5% of the 163 cells used in the cluster analysis), where plateaus and spiking episodes alternate, are observed at the boundary of RS and PP clusters. This suggested to us that a conductance-based model incorporating only the leak current, $I_{Nat}$, $I_{Nap}$, and $I_{Kdr}$ (see Materials and methods), could account for most experimental observations, the observed zonation being explained in terms of

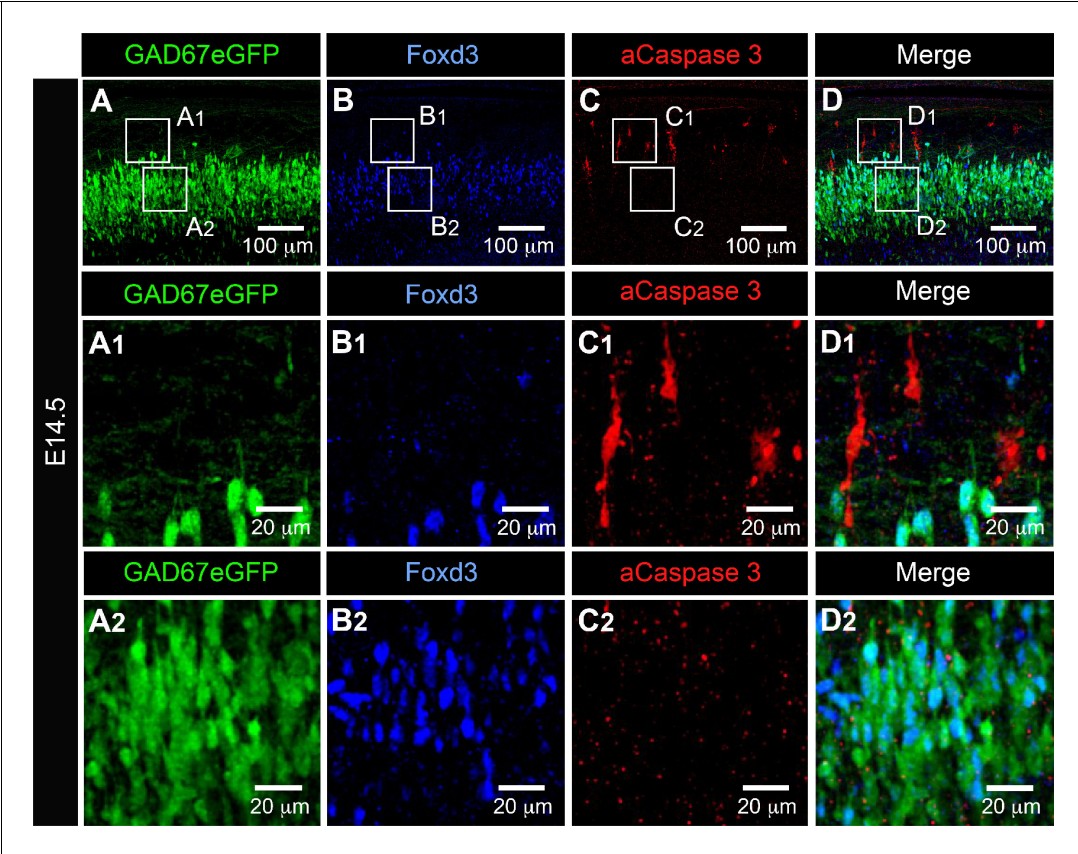

**Figure 5.** Activated caspase3 (aCaspase3) is not observed in embryonic V1$^R$ at E14.5. Representative confocal image of the ventral part of an isolated lumbar spinal cord of E14.5 GAD67-eGFP mouse embryo showing immunostainings using antibodies against eGFP (**A**), FoxD3 (**B**), and aCaspase3 (**C**). (**D**) Superimposition of the three stainings shows that embryonic V1$^R$ (eGFP+ and FoxD3+) were not aCaspase three immunoreactive. (**A1, B1, C1,** and **D1**). Enlarged images from (**A**), (**B**), and (**C**) showing that aCaspase3 staining is localized in areas where eGFP and Foxd3 staining were absent. (**A2, B2, C2,** and **D2**) Enlarged images from (**A**), (**B**), and (**C**) showing that aCaspase3 staining is absent in the area where V1$^R$ (eGFP+ and FoxD3+) are located. aCaspase3 staining that did not co-localize with GAD67eGFP likely indicates motoneuron developmental cell death.

bifurcations between the different stable states of the model. Therefore, we first investigated a simplified version of the model without $I_A$ and slow inactivation of $I_{Nap}$.

A one-parameter bifurcation diagram of this 'basic' model is shown in *Figure 7B* for two values of $G_{Kdr}$ (2.5 nS and 10 nS) and a constant injected current $I$ = 20 pA. In both cases, the steady-state membrane voltage (stable or unstable) and the peak and trough voltages of stable and unstable periodic solutions are shown as a function of the maximal conductance $G_{Nap}$ of the $I_{Nap}$ current, all other parameters being kept constant. For $G_{Kdr}$ = 10 nS, the steady-state membrane voltage progressively increases (in gray) with $G_{Nap}$, but RS (in red, see voltage trace for $G_{Nap}$ = 1.2 nS) is not achieved until $G_{Nap}$ reaches point SN$_1$, where a saddle node (SN) bifurcation of limit cycles occurs. This fits with the experimental data, where a minimal value of $G_{Nap}$ is required for RS (see also *Boeri et al., 2018*), and is in agreement with the known role of $I_{Nap}$ in promoting repetitive discharge (*Taddese and Bean, 2002*; *Kuo et al., 2006*). Below SN$_1$, the model responds to the onset of a current pulse by firing only one spike before returning to quiescence (see voltage trace for $G_{Nap}$ = 0.2 nS) or a few spikes when close to SN$_1$ (not shown) before returning to quiescence. The quiescent state becomes unstable through a subcritical Hopf bifurcation (HB) at point HB$_1$, with bistability between quiescence and spiking occurring between SN$_1$ and HB$_1$ points. Repetitive firing persists when $G_{Nap}$ is increased further and eventually disappears at point SN$_2$. The firing rate does not increase much throughout the RS range (*Figure 7—figure supplement 1C*), remaining between 11.5 Hz (at SN$_1$) and 20.1 Hz (at SN$_2$). A stable plateau appears at point HB$_2$ through a subcritical

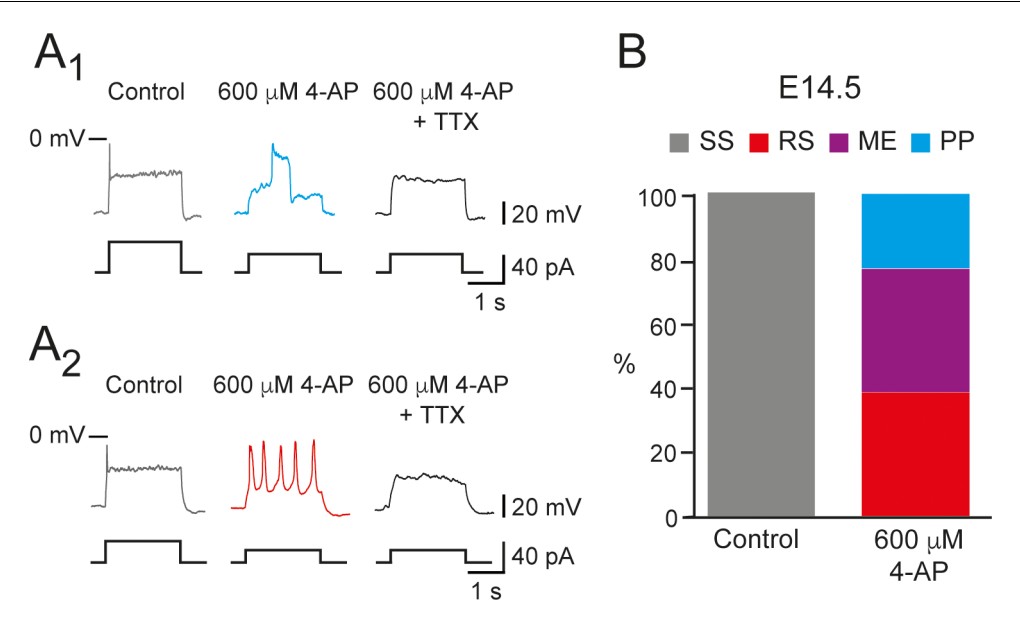

**Figure 6.** 600 μM 4-aminopiridine (4-AP) changed the firing pattern of single spiking (SS) embryonic V1$^R$ recorded at E14.5. The firing pattern of embryonic V1$^R$ was evoked by 2 s suprathreshold depolarizing current steps. (**A**) Representative traces showing the effect of 4-AP application (600 μM) on the firing pattern of SS V1$^R$ recorded at E14.5. Note that the applications of 600 μM 4-AP evoked either a plateau potential (PP, **A1**) or repetitive spiking (RS, **A2**), both fully blocked by tetrodotoxin. (**B**) Bar plots showing the proportions of the different firing patterns observed in the presence of 600 μM 4-AP versus control recorded in SS V1$^R$ at E14.5 (n = 14; N = 14). SS V1$^R$ (gray), RS V1$^R$ (red), mixed Events (ME) V1$^R$ (purple), and PP V1$^R$ (blue).

The online version of this article includes the following figure supplement(s) for figure 6:

**Figure supplement 1.** $I_{Nap}$ is present in embryonic V1$^R$ recorded at E14.5.

**Figure supplement 2.** $I_{Kdr}$ was inhibited by 4-aminopiridine (4-AP) in V1$^R$ recorded at E14.5.

HB. The model is bistable between HB$_2$ and SN$_2$, with plateau and large amplitude APs coexisting in this range.

The model behaves very differently when $G_{Kdr}$ is reduced to 2.5 nS (gray-blue curve in *Figure 7B*). It exhibits a unique stable fixed point whatever the value of $G_{Nap}$ is, and the transition from quiescence to plateau is gradual as $G_{Nap}$ is increased. No RS is ever observed. This indicates that the activity pattern is controlled not only by $G_{Nap}$ but also by $G_{Kdr}$. This is demonstrated further in *Figure 7C*, where $G_{Nap}$ was fixed at 1.2 nS while $G_{Kdr}$ was increased from 0 to 25 nS. The model exhibits a PP until $G_{Kdr}$ is increased past point the subcritical HB point HB$_2$, RS sets in before at point SN$_2$ via a SN of limit cycles bifurcation. When $G_{Kdr}$ is further increased, repetitive firing eventually disappears through a SN bifurcation of limit cycles at point SN$_1$, the quiescent state becomes stable through a subcritical HB at point HB$_1$, and bistability occurs between these two points. This behavior is in agreement with *Figure 7A*.

Since both conductances $G_{Nap}$ and $G_{Kdr}$ control the firing pattern of embryonic V1$^R$ cells, we computed a two-parameter bifurcation diagram (*Figure 7D*), where the stability regions of the different possible activity states and the transition lines between them are plotted in the $G_{Nap}$ - $G_{Kdr}$ plane. The black curves correspond to the bifurcations HB$_1$ and HB$_2$ and delimit a region where only repetitive firing occurs. The red curves correspond to the SN bifurcations of periodic orbits associated with the transition from quiescence to firing (SN$_1$) and the transition from plateau to firing (SN$_2$). They encompass a region (shaded area) where RS can be achieved but may coexist with SS (between the HB$_1$ and SN$_1$ lines) or PP (in the narrow region between the HB$_2$ and SN$_2$ lines).

Some important features of the diagram must be emphasized: (1) minimal values of both $G_{Nap}$ (to ensure sufficient excitability) and $G_{Kdr}$ (to ensure proper spike repolarization) are required for RS; (2) SS and PP can be clearly distinguished only when they are separated by a region of RS (see also

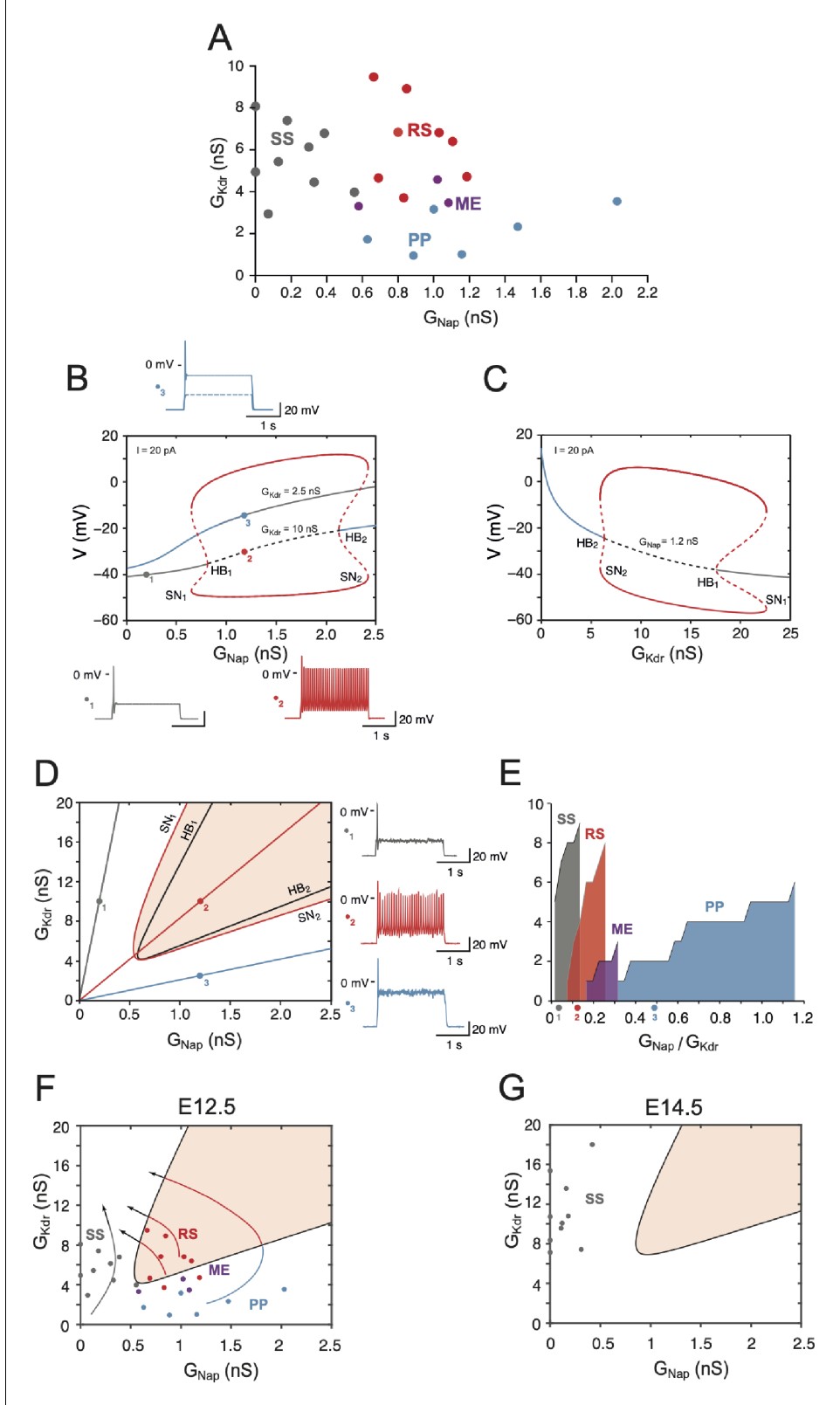

**Figure 7.** Embryonic V1$^R$ firing patterns predicted by computational modeling. (**A**) Firing patterns of 26 recorded cells, in which both $G_{Nap}$ and $G_{Kdr}$ were measured. Gray: single spiking (SS); red: repetitive spiking (RS); blue: plateau potential (PP). The three purple points located at the boundary between the RS and PP regions correspond to mixed events (ME), where plateau potentials alternate with spiking episodes. Note that no cell
*Figure 7 continued on next page*

*Figure 7 continued*

exhibited low values of both $G_{Nap}$ and $G_{Kdr}$ (lower left) or large values of both conductances (upper right). (B) Bifurcation diagram of the deterministic model when $G_{Kdr}$ is kept fixed to 2.5 nS or 10 nS while $G_{Nap}$ is varied between 0 and 2.5 nS. $G_{in}$ = 1 nS and $I$ = 20 pA. For $G_{Kdr}$ = 10 nS (i.e., in the top experimental range), the red curves indicate the maximal and minimal voltages achieved on the stable limit cycle associated with repetitive firing (solid lines) and on the unstable limit cycle (dashed lines). The fixed point of the model is indicated by a gray solid line when it corresponds to the stable quiescent state, a gray dashed line when it is unstable, and a solid blue line when it corresponds to a stable plateau potential. The two Hopf bifurcations (HB) corresponding to the change of stability of the quiescence state (HB$_1$, $G_{Nap}$ = 0.81 nS) and the voltage plateau (HB$_2$, $G_{Nap}$ = = 2.13 nS) are indicated, as well as the two saddle node (SN) bifurcations of limit cycles associated with the onset (SN$_1$, $G_{Nap}$ 0.65 nS) and offset (SN$_2$, $G_{Nap}$ = 2.42 nS) of repetitive spiking as $G_{Nap}$ is increased. For $G_{Kdr}$ = 2.5 nS, the model does not display repetitive firing; it possesses a unique fixed point, which is always stable (blue-gray curve). The transition from quiescence to plateau is gradual with no intervening bifurcation. Representative voltage traces of the three different activity patterns are shown: SS in response to a 2 s current pulse (gray, $G_{Nap}$= 0.2 nS, $G_{Kdr}$= 10 nS), RS (red, $G_{Nap}$= 1.2 nS, $G_{Kdr}$= 10 nS), and PP (blue, $G_{Nap}$= 1.2 nS, $G_{Kdr}$= 2.5 nS). Note that the plateau never outlasts the current pulse. (C) Bifurcation diagram when $G_{Nap}$ is kept fixed at 1.2 nS and $G_{Kdr}$ is varied between 0 and 25 nS ($I$ = 20 pA). Same conventions as in (B). PP is stable until the subcritical HB$_2$ ($G_{Kdr}$ = 6.34 nS) is reached, repetitive firing can be observed between SN$_2$ ($G_{Kdr}$ = 5.93 nS) and SN$_1$ ($G_{Kdr}$ = = 22.65 nS). The quiescent state is stable from point HB$_1$ ($G_{Kdr}$= 17.59 nS) onward. (D) Two-parameter bifurcation diagram of the model in the $G_{Nap}$ - $G_{Kdr}$ plane ($I$ = 20 pA). The black curves indicate the bifurcations HB$_1$ and HB$_2$. The red curves indicate the SN bifurcations of limit cycles SN$_1$ and SN$_2$. The shaded area indicates the region where repetitive firing can occur. The oblique lines through the points labeled 1, 2, and 3, the same as in (B), correspond to three different values of the ratio of $G_{Nap}$ / $G_{Kdr}$: 0.02 (gray), 0.12 (red), and 0.48 (blue). Voltage traces on the right display the response to a 2 s current pulse when channel noise is taken into account for the three regimes: SS (top, gray trace and dot in the diagram), RS (middle, red), and PP (bottom, blue). They correspond to the three deterministic voltage traces shown in (B). Note that the one-parameter bifurcation diagrams shown in (B) correspond to horizontal lines through points 1 and 2 ($G_{Kdr}$ = 10 nS) and through point 3 ($G_{Kdr}$ = 2.5 nS), respectively. The bifurcation diagram in (C) corresponds to a vertical line through points 2 and 3 ($G_{Nap}$ = 1.2 nS). (E) Cumulative distribution function of the ratio $G_{Nap}/G_{Kdr}$ for the four clusters in (A), showing the sequencing SS (gray) → RS (red) → ME (purple, three cells only) → PP (blue) predicted by the two-parameter bifurcation diagram in (D). The wide PP range, as compared to SS and RS, merely comes from the fact that $G_{Nap}$ is small for cells in this cluster. The three colored points indicate the slopes of the oblique lines displayed in (D) . (F) The data points in (A) are superimposed on the two-parameter bifurcation diagram shown in (D), demonstrating a good agreement between our basic model and experimental data (same color code as in (A) for the different clusters). The bifurcation diagram is simplified compared to (A), only the region where repetitive spiking is possible (i.e., between the lines SN$_1$ and SN$_2$ in A) being displayed (shaded area). Notice that three ME cells (purple dots) are located close to the transition between the RS and PP regions. The four arrows indicate the presumable evolution of $G_{Nap}$ and $G_{Kdr}$ for SS, RS, ME, and PP cells between E12.5 and E14.5–15.5. $G_{Nap}$ eventually decreases while $G_{Kdr}$ keeps on increasing. (G) Distribution of a sample of cells in the $G_{Nap}$ - $G_{Kdr}$ plane at E14.5. All the cells are located well within the SS region far from bifurcation lines because of the decreased $G_{Nap}$ compared to E12.5, the increased $G_{Kdr}$, and the shift of the RS region (shaded) due to capacitance increase (18 versus 13 pF).

The online version of this article includes the following figure supplement(s) for figure 7:

**Figure supplement 1.** Effect of $I_A$ on embryonic V1$^R$ firing patterns predicted by computational modeling.

**Figure supplement 2.** Explaining the effect of 4-aminopiridine (4-AP) on the firing pattern.

---

*Figure 7B* for $G_{Kdr}$ = 10 nS), otherwise the transition is gradual (*Figure 7B* for $G_{Kdr}$ = 2.5 nS); and (3) only oblique lines with an intermediate slope cross the bifurcation curve and enter the RS region (e. g., see the red line in *Figure 7D*). This means that RS requires an appropriate balance between $I_{Nap}$ and $I_{Kdr}$. If the ratio $G_{Nap}/G_{Kdr}$ is too large (blue line) or too small (gray line), only PPs or SS will be observed at steady state. This is exactly what is observed in experiments, as shown by the cumulative distribution function of the ratio $G_{Nap}/G_{Kdr}$ for the different clusters of embryonic V1$^R$ in *Figure 7E* (same cells as in *Figure 7A*). The ratio increases according to the sequence SS → RS → ME → PP, with an overlap of the distributions for SS V1$^R$ and RS V1$^R$. Note also that the ratio for ME cells (around 0.25) corresponds to the transition between RS and PP (more on this below).

Embryonic V1$^R$ cells display voltage fluctuations that may exceed 5 mV and are presumably due to channel noise. The relatively low number of sodium and potassium channels (of the order of a few thousands) led to voltage fluctuations in the stochastic version of our model comparable to those

seen experimentally when the cell was quiescent (top voltage trace in *Figure 7D*) or when a voltage plateau occurred (bottom trace). Channel noise caused some jitter during RS (middle trace) and induced clearly visible variations in the amplitude of APs. However, RS proved to be very robust and was not disrupted by voltage fluctuations. Altogether, channel noise little alters the dynamics (compare the deterministic voltage traces in *Figure 7B* and the noisy traces in *Figure 7D*). This is likely because channel noise has a broad power spectrum and displays no resonance with the deterministic solutions of the model.

The one-parameter bifurcation diagram of the model was not substantially modified when we took $I_A$ into account, as shown in *Figure 7—figure supplement 1*. It just elicited a slight membrane hyperpolarization, an increase in the minimal value of $G_{Nap}$ required for firing, and a decrease of the firing frequency. The transition from repetitive firing to plateau was not affected because $I_A$ is then inactivated by depolarization.

The bifurcation diagram of *Figure 7D* accounts *qualitatively* for the physiological data on V1$^R$ at E12.5 presented in *Figure 7A*, as shown in *Figure 7F* where the conductance data of *Figure 7A* were superimposed on it. However, one must beware of making a more *quantitative* comparison because the theoretical bifurcation diagram was established for a constant injected current of 20 pA, whereas the current injected in experiments data varied from neuron to neuron and ranged from 10 to 30 pA in the sample shown in *Figure 7A*. The position of bifurcation lines in the $G_{Nap}$ - $G_{Kdr}$ plane depends not only on the value of the injected current, but on the values chosen for the other parameters, which also vary from cell to cell but were kept at fixed values in the model (*Ori et al., 2018*). For instance, the diagrams were computed in *Figure 7D, F* for $G_{in}$ = 1 nS and $C_{in}$ = 13 pF, the median values of the input conductance and capacitance at E12.5, taking no account of the cell-to-cell variations of these quantities. Between E12.5 and E14.5, $C_{in}$, which provides an estimate of the cell size, increases by 38% in average, whereas $G_{in}$ is not significantly modified (see *Figure 4*). As illustrated in *Figure 7G*, the two-parameter bifurcation diagram is then shifted upward and rightward compared to *Figure 7F* because larger conductances are required to obtain the same firing pattern. The observed regression of excitability from E12.5 to E14.5–E15.5 (see *Figure 4C*) thus comes from a decrease in $G_{Nap}$ density (see presumable developmental trajectories indicated by arrows in *Figure 7F*) together with a shift of the RS region as cell size increases. As a result, all 10 cells shown in *Figure 7G* are deeply inside the SS region at E14.5.

It is less straightforward to explain on the basis of our model the experiments where 4-AP changed the firing pattern of SS V1$^R$ (*Figure 2*). Indeed, the decrease of $G_{Kdr}$ (*Figure 7—figure supplement 2*), although it may exceed 70% at the higher concentrations of 4-AP we used, is not sufficient by itself to account for the change in the firing pattern of V1$^R$ because data points in the SS cluster will not cross the bifurcation lines between SS and RS (SN$_1$) and between RS and PP (SN$_2$) when displaced downward in the $G_{Nap}$ - $G_{Kdr}$ plane. However, 4-AP at a 300 μM concentration also decreased $G_{in}$ (by 23% in average and up to 50% in some neurons), the rheobase current with it, and the current that was injected in cells during experiments was reduced accordingly. When we take into account this reduction of both $G_{in}$ and $I$, the two-parameter bifurcation diagram of the model remains qualitatively the same, but it is shifted leftwards and downwards in the $G_{Nap}$ - $G_{Kdr}$ plane (*Figure 7—figure supplement 2*). As a consequence, the bifurcation lines between SS and RS (SN1) and between RS and PP (SN$_2$) are then successively crossed when $G_{Kdr}$ is reduced, in accordance with experimental results.

## Theoretical analysis: slow inactivation of I$_{Nap}$ and bursting

Our basic model accounts for the firing pattern of 73% of the 163 cells used in the cluster analysis. However, bursting, under the form of recurring plateaus separated by brief repolarization episodes (see a typical trace in *Figure 8A*, left), was experimentally observed in half of PP V1$^R$ (24 out of 46), and plateaus intertwined with spiking episodes were recorded in the 13 cells of the ME cluster (8% of the total sample, see *Figure 8A, right*, for a typical example). Recurrent plateaus indicate membrane bistability and require that the $I - V$ curve be S-shaped. This occurs when $G_{Nap}$ is large and $G_{Kdr}$ small (*Figure 8B1, B2*). However, our basic model lacks a mechanism for switching between quiescent state and plateau, even in this case. Channel noise might induce such transitions, but our numerical simulations showed that this is too infrequent to account for bursting (see voltage trace in *Figure 8B1* where the plateau state is maintained despite channel noise).

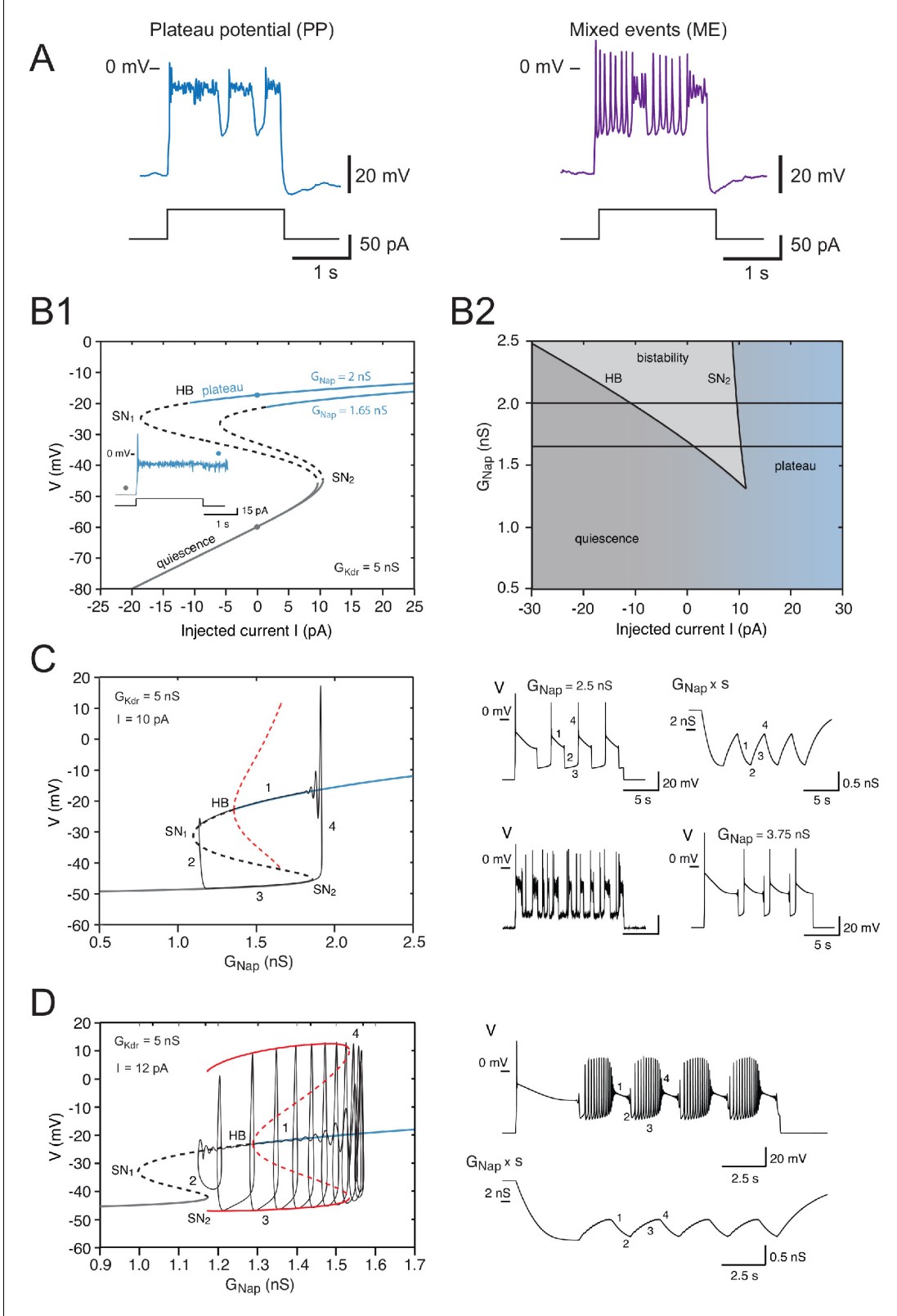

**Figure 8.** Effects of the slow inactivation of $I_{Nap}$ on firing patterns predicted by computational modeling. (**A**) Examples of repetitive plateaus (left) and mixed events (right) recorded in V1R at E12.5 during a 2 s current pulse. (**B1**) Current-voltage curve of the basic model (without slow inactivation of $I_{Nap}$ and without $I_A$ or channel noise) for $G_{Kdr}$ = 5 nS and for $G_{Nap}$ = 1.65 nS (lower curve) and 2 nS (upper curve). Solid lines denote stable fixed points and dashed lines unstable ones. For $G_{Nap}$ = 1.65 nS, bistability between quiescence and plateau occurs between 1.39 and 10.48 pA. When $G_{Nap}$ is increased

*Figure 8 continued on next page*

*Figure 8 continued*

to 2 nS, the bistability region ranges from –10.84 to 9.70 pA, thus extending into the negative current range. This implies that once a plateau has been elicited, the model will stay in that stable state and not return to the resting state, even though current injection is switched off (see inset). (**B1** inset) Voltage response to a 2 s current pulse of 15 pA for $G_{Nap}$ = 2 nS. The resting state (gray dot on the lower curve in **B1**) is destabilized at pulse onset and a plateau is elicited (blue dot on the upper curve in **B1**). At pulse offset, the plateau is maintained, even though the injected current is brought back to zero, and channel noise is not sufficient to go back to the resting state. (**B2**) Domain of bistability between quiescence and plateau (shaded) in the $I - G_{Nap}$ plane for $G_{Kdr}$ = 5 nS. It is delimited by the line $SN_2$ where a SN bifurcation of fixed points occurs and by the subcritical Hopf bifurcation line HB where the plateau becomes unstable. Bistability requires that $G_{Nap}$ exceeds 1.35 nS, and the domain of bistability enlarges as $G_{Nap}$ is increased further. The two horizontal lines correspond to the two cases shown in (**B1**) $G_{Nap}$ = 1.65 nS and 2 nS. (**C**) Behavior of the model when slow inactivation is incorporated. The bifurcation diagram of the basic model (without slow inactivation) for $I$ = 10 pA and $G_{Kdr}$ = 5 nS (same conventions as in *Figure 7B*) and the stable limit cycle (black solid curve) obtained when slow inactivation is added are superimposed. The limit cycle is comprised of four successive phases (see labels): (1) long plateau during which $I_{Nap}$ slowly inactivates, (2) fast transition to the quiescent state, (3) repolarization episode during which $I_{Nap}$ slowly de-inactivates, and (4) fast transition back to the plateau. Each plateau starts with a full-blown action potential followed by rapidly decaying spikelets. Note that the bifurcation HB is subcritical here (unstable limit cycle shown by dashed red curve), at variance with square wave bursting (supercritical bifurcation and stable limit cycle); this is a characteristic feature of pseudo-plateau bursting. Note also that the plateau extends beyond the bifurcation HB because it is only weakly unstable then. Responses to a 15 s current pulse are shown on the right side. Top left: voltage response ($G_{Nap}$ = 2.5 nS); top right: behavior of the 'effective' conductance of the $G_{Nap}$ channels, that is, the maximal conductance $G_{Nap}$ multiplied by the slow inactivation variable $s$. Bottom left: voltage trace when channel noise is added to fast and slow gating variables; bottom right: voltage trace when $G_{Nap}$ is increased by 50% to 3.75 nS. (**D**) Mixed events. The bifurcation diagram of the basic model for $G_{Kdr}$ = 5 nS and $I$ = 12 pA and the stable limit cycle obtained in the presence of slow inactivation ($G_{Nap}$ = 2.5 nS) are superimposed. Here again, the limit cycle comprises four successive phases (see labels): (1) slow inactivation of $I_{Nap}$ that leads to the crossing of the bifurcation point $HB_2$ and then to the destabilization of the plateau potential; (2) fast transition to the spiking regime; (3) repetitive spiking during which $I_{Nap}$ slowly de-inactivates, which leads to the crossing of the bifurcation point $SN_2$ and terminates the spiking episode; and (4) fast transition back to the stable plateau potential. Response to a 15 s current pulse of 12 pA is shown on the right in the absence of any channel noise. Top: voltage trace (same labels as in the bifurcation diagram on the left); bottom: variations of the 'effective' conductance $G_{Nap}s$ (same labels as in the voltage trace). Note that de-inactivation sufficient to trigger a new plateau occurs over a series of successive spikes, hence the small oscillations visible on the trace. Note also that in (**C**) and (**D**) the first plateau lasts longer than the following ones, as in electrophysiological recordings of embryonic V1$^R$ cells displaying repetitive plateaus. This form of adaptation is caused by the slow inactivation of the persistent sodium current.

To explain recurrent plateaus during a constant current pulse, we have to incorporate in our model an additional slow dynamical process. Therefore, we took into account the slow inactivation of $I_{Nap}$ that is observed in experiments. $I_{Kdr}$ also inactivates slowly but over times that are much longer than the timescale of bursting, which is why we did not take its slow inactivation into account. The one-parameter bifurcation diagram of the basic model without slow inactivation of $I_{Nap}$ is shown in *Figure 8C* for $G_{Kdr}$ = 5 nS and an injected current reduced to 10 pA (as compared to 20 pA in the previous section), so as to allow for bistability (see *Figure 8B2*). The $G_{Nap} - V$ curve is then S-shaped, as shown in *Figure 8B1*, with a bistability region for $G_{Nap}$ between 1.36 and 1.85 nS. This is in contrast with *Figure 7B* where the $G_{Nap} - V$ curve was monotonic. Adding the slow (de)inactivation of $I_{Nap}$ then causes periodic transitions between up (plateau) and down (quiescent) states, as illustrated by the top voltage trace on the right of *Figure 8C*, and the model displayed a stable limit cycle (shown in black in the bifurcation diagram on the left of *Figure 8C*). This mechanism is known as pseudo-plateau or plateau-like bursting (a.k.a. fold-subcritical HB bursting) (*Teka et al., 2011*). In contrast with square wave bursting (*Bertram et al., 1995*; *Izhikevich, 2000a*; *Borisyuk and Rinzel, 2005*; *Rinzel, 1985*), where the up state is a stable limit cycle arising from a supercritical Hopf bifurcation (*Stern et al., 2008*; *Osinga and Tsaneva-Atanasova, 2010*; *Osinga and Tsaneva-Atanasova, 2010*; *Osinga et al., 2012*), the up state here is a stable fixed point (which coexists with an unstable limit cycle). This is why one does not observe bursts of APs separated by quiescent periods as, for instance, observed in postnatal CA1 pyramidal cells (*Golomb et al., 2006*) and in neurons of neonatal pre-Bötzinger Complex (*Del Negro et al., 2002*; *Rybak et al., 2004*), but recurrent plateaus. The duration of the plateaus and repolarization episodes depends on the values of $G_{Nap}$ and $G_{Kdr}$. A voltage-independent time constant $\tau_s$= 2 ms leads to up and down states of comparable durations (see top-left voltage trace in *Figure 8C*). In agreement with the bifurcation diagram of *Figure 8C*, the persistent sodium current inactivates during plateaus (phase 1, see top-right trace in *Figure 8C*) and de-inactivates during quiescent episodes (phase 3, see top-right trace). Transitions from the up state to the down state occurs when inactivation is maximal (phase 2) and transition from the down state to the up state when it is minimal (phase 4). Adding channel noise preserves bursting but introduces

substantial randomness in the duration of plateaus and repolarization episodes (bottom-left voltage trace in *Figure 8C*). Moreover, it substantially decreases the duration of both plateaus and quiescent episodes by making transition between the two states easier (compare the top and bottom voltage traces on the left, both computed for $\tau_s = 2ms$).

Increasing $G_{Nap}$ (or decreasing $G_{Kdr}$) makes plateaus much longer than quiescent episodes (see bottom-right voltage trace in *Figure 8C*). This again points out to the fact that the ratio of these two conductances is an important control parameter. We also noted that adding the $I_A$ current lengthened the quiescent episodes (*Figure 7—figure supplement 1*).

Slow inactivation of $I_{Nap}$ also provides an explanation for mixed patterns, where plateaus alternate with spiking episodes (*Figure 8A, right*). They take place in our model near the transition between RS and PP, as in experiments (see *Figure 8A*). Slow inactivation can lead to elliptic bursting, notably when the bifurcation HB is subcritical (*Izhikevich, 2000b*; *Su et al., 2004*), which is the case here (*Figure 8D*). The model then displays a stable limit cycle with alternating plateaus and spiking episodes, arising from crossing the bifurcation points $HB_2$ and $SN_2$ back and forth (see bifurcation diagram in *Figure 8D* and top voltage trace). We note that sufficient de-inactivation of $I_{Nap}$ for triggering a new plateau (phase 1 in the bottom trace of *Figure 8D*) may be difficult to achieve during spiking episodes because voltage oscillates over a large range, which tends to average out the variations of the inactivation level. If de-inactivation is not sufficient, the model keeps on spiking repetitively without returning to the plateau state. This is what occurs for cells well within the RS region, far away from the RS-PP transition. It also probably explains why it was difficult in many recorded cells to elicit plateaus by increasing the injected current, activation of $I_{Nap}$ induced by the larger current being balanced by the increased inactivation.

Altogether, our study shows that a model incorporating the slow inactivation of $I_{Nap}$ accounts for all the firing patterns displayed by cells from the PP and ME clusters.

## Discussion

V1$^R$ constitute a homogeneous population when referring to their transcription factor program during development (*Benito-Gonzalez and Alvarez, 2012*; *Stam et al., 2012*), their physiological function (*Eccles et al., 1956*), and their firing pattern at postnatal stages (*Bikoff et al., 2016*). Surprisingly, our electrophysiological recordings and our cluster analysis clearly indicate that distinct functional classes of V1$^R$ are transiently present during development at the onset of the SNA (E11.5–E12.5). Five different groups of embryonic V1$^R$ were defined using cluster analysis, according to their firing properties.

### Development of the firing pattern of embryonic V1$^R$ during SNA

It is generally assumed that, during early development, newborn neurons cannot sustain repetitive firing (*Pineda and Ribera, 2010*; *Spitzer et al., 2000*). Later on, neurons progressively acquire the ability to fire repetitively, APs become sharper, and neurons eventually reach their mature firing pattern due to the progressive appearance of a panoply of voltage-gated channels with different kinetics (*Moody and Bosma, 2005*; *Pineda and Ribera, 2010*; *Spitzer et al., 2000*). Our results challenge the general view that SS is a more primitive form of excitability (*Pineda and Ribera, 2010*). Indeed, we show that repetitive firing and PPs dominated at early stages (E11.5–E12.5), while SS was prevailing only later (E13.5–E16.5).

The different V1$^R$ firing patterns observed at E11.5–E12.5 might reflect variability in the maturation level between V1$^R$ at a given developmental stage, as suggested for developing MNs (*Vinay et al., 2000*; *Durand et al., 2015*). However, this is unlikely since V1$^R$ transiently lose their ability to sustain tonic firing or PP after E13.5. The heterogeneous discharge patterns of V1$^R$ observed before E13.5 contrast with the unique firing pattern of V1$^R$ at postnatal age (*Bikoff et al., 2016*). Accordingly, the transient functional heterogeneity of V1$^R$ rather reflects an early initial developmental stage (E11.5–E13.5) of intrinsic excitability.

The physiological meaning of the transient functional involution of V1$^R$ that follows, after E12.5, is puzzling. To our knowledge, such a phenomenon was never described in vertebrates during CNS development. So far, a functional involution was described only for inner hair cells between E16 and P12 (*Marcotti et al., 2003a*; *Marcotti et al., 2003b*) and cultured oligodendrocytes (*Sontheimer et al., 1989*), and it was irreversible. Because most V1$^R$ cannot sustain tonic firing after

E12.5, it is likely that their participation to SNA is limited to the developmental period before other GABAergic IN subtypes mature and start to produce GABA and glycine (*Allain et al., 2004*). Interestingly, embryonic V1$^R$ begin to recover their capability to sustain tonic firing when locomotor-like activity emerges (*Myers et al., 2005*; *Yvert et al., 2004*), a few days before they form their recurrent synaptic loop with MNs (around E18.5 in the mouse embryos; *Sapir et al., 2004*). One possible function of the transient involution between E12.5 and E15.5 could be to regulate the growth of V1$^R$ axons toward their targets. It is indeed known that low calcium fluctuations within growth cones are required for axon growth while high calcium fluctuations stop axon growth and promote growth cone differentiation (*Henley and Poo, 2004*).

## Ion channels mechanisms underlying the functional heterogeneity of embryonic V1$^R$

Blockade of $I_{Nap}$ leads to single spiking (*Boeri et al., 2018*), which emphasizes the importance of this current for the occurrence of repetitive firing and plateau potentials in V1$^R$ at early developmental stages. But these neurons can also switch from one firing pattern to another, when $G_{Kdr}$ is decreased by 4-AP, which emphasizes the importance of $I_{Kdr}$. We found that the main determinant of embryonic V1$^R$ firing pattern is the balance between $G_{Nap}$ and $G_{Kdr}$.

A Hodgkin–Huxley-type model incorporating a persistent sodium current $I_{Nap}$ provided a parsimonious explanation of all five firing patterns recorded in the V1$^R$ population at E12.5. It provided a mathematical interpretation for the clustering of embryonic V1$^R$ shown by the hierarchical analysis and accounted for the effect of 4-AP and riluzole (*Boeri et al., 2018*) on the discharge. Remarkably, it highlighted how a simple mechanism involving only the two opposing currents $I_{Nap}$ and $I_{Kdr}$, could produce functional diversity in a population of developing neurons. The model explained why minimal $G_{Nap}$ and $G_{Kdr}$ are required for firing, and how a synergy between $G_{Nap}$ and $G_{Kdr}$ controls the firing pattern and accounts for the zonation of the $G_{Nap} - G_{Kdr}$ plane that is observed experimentally.

Taking into account the slow inactivation of $I_{Nap}$ to the model allowed us to explain the bursting patterns displayed by cells of the PP and ME clusters. We showed, in particular, that smooth repetitive plateaus could be explained by a pseudo-plateau bursting mechanism (*Teka et al., 2011*; *Osinga and Tsaneva-Atanasova, 2010*). Such bursting scenario has been previously studied in models of endocrine cells (*Stern et al., 2008*; *Tsaneva-Atanasova et al., 2010*; *Tagliavini et al., 2016*) and adult neurons (*Oster et al., 2015*), but rarely observed in experiments (*Chevalier et al., 2016*). It contrasts with the more common square wave bursting at firing onset, that is, alternating bursts of APs and quiescent episodes, on which most studies of bursting focused (*Golomb et al., 2006*; *Del Negro et al., 2002*; *Rybak et al., 2004*). Our model can also display such square wave bursting, but this occurs for physiologically unrealistic parameter values, so we did not dwell on that bursting mode that we never observed in embryonic V1$^R$. The model also provides a mathematical explanation for mixed events, where bursts of APs alternate with plateau episodes. It is due to an elliptic bursting scenario at the RS-PP transition, a firing range that the aforementioned studies did not examine. This further emphasizes the capacity of our simple model to account for a wide diversity of firing patterns.

Pseudo-plateau bursting has also been observed in the embryonic pre-Bötzinger network (*Chevalier et al., 2016*). However, it is produced there by the calcium-activated nonselective cationic current $I_{CAN}$, while $I_{Nap}$ leads to square wave bursting. Pseudo-plateau bursting, displayed by half of the cells at E16.5, largely disappears at E18.5 because of the change in the balance between $I_{CAN}$ and $I_{Nap}$ during embryonic maturation (*Chevalier et al., 2016*). Such a scenario cannot account for the variety of discharge patterns observed in embryonic V1$^R$ at the E11.5–12.5 stage of development. Our theoretical analysis and experimental data clearly indicate that the interplay between two opposing currents is necessary to explain all the firing patterns of V1$^R$. Our model is of course not restricted to embryonic V1$^R$, but may also apply to any electrically compact cell, the firing activity of which is dominated by $I_{Nap}$ and delayed rectifier potassium currents. This is the case of many classes of embryonic cells in mammals at an early stage of their development. It can also apply to the axon initial segment, where $G_{Nap}$ and $G_{Kdr}$ are known to play the major role in the occurrence of repetitive firing (*Kole and Stuart, 2012*).

Altogether, our experimental and theoretical results provide a global view of the developmental trajectories of embryonic V1$^R$ (see *Figure 7F, G*). At E12.5, conductances of embryonic V1$^R$ are

widely spread in the $G_{Nap} - G_{Kdr}$ plane, which explains the heterogeneity of their firing patterns. This likely results from the random and uncorrelated expression of sodium and potassium channels from cell to cell at this early stage. Between E12.5 and E14.5–15.5, cell size increases, and $G_{Kdr}$ with it, while the density of sodium channels decreases (see *Figures 1* and *4*). The functional involution displayed by V1$^R$ between E12.5 and E15.5 thus mainly results from a decrease of $G_{Nap}$ coordinated with an increase of $G_{Kdr}$. How these synergistic processes are controlled during this developmental period remains an open issue.

It is important to note that the presence of $I_{Nap}$ is required for the functional diversity of V1$^R$. Indeed, in the absence of $I_{Nap}$, V1$^R$ lose their ability to generate plateau potentials or to fire repetitively. More generally, when the diversity of voltage-gated channels is limited, as observed in embryonic neurons (*Moody and Bosma, 2005*), changes in the balance between $I_{Kdr}$ and non inactivating inward currents can modify the firing pattern. This can be achieved not only by $I_{Nap}$, but also by other slowly or non-inactivating inward conductances, such as $I_{CAN}$ (*Chevalier et al., 2016*). Our work also clearly indicates that a change in the firing pattern can only occur if a change in inward conductances cannot be counterbalanced by a corresponding change in outward conductances. This implies that there is no homeostatic regulation of channel density to ensure the robustness of V1$^R$ excitability during its early development, contrarily to the mature CNS (*O'Leary et al., 2013*). In addition, the poor repertoire of voltage-gated channels at this developmental stage precludes channel degeneracy, which is also known to ensure the robustness of excitability in mature neurons (*O'Leary et al., 2013*).

In conclusion, our study shows that there is no universal pattern of development in embryonic neurons, and it demonstrates that a simple general mechanism involving only two slowly inactivating voltage-gated channels with opposite effects is sufficient to produce a wide variety of firing patterns in immature neurons having a limited repertoire of voltage-gated channels.

# Materials and methods

## Key resources table

| Reagent type (species) or resource | Designation | Source or reference | Identifiers | Additional information |
|---|---|---|---|---|
| Genetic reagent (*Mus musculus* Swiss) male and female | GAD1$^{GFP}$ | PMID:14574680 | | A cDNA encoding enhanced GFP (eGFP) was targeted to the locus encoding the gene Gad1 |
| Antibody | Anti-FoxD3 (Guinea pig polyclonal) | PMID:19088088 | | IF(1:5000) |
| Antibody | Anti-cleaved Caspase-3 (Asp175) (Rabbit polyclonal) | Cell Signaling Technology | Cat# 9661, RRID:AB_2341188 | IF(1:1000) |
| Chemical compound, drug | Tetrodotoxin | Alomone Labs | Cat# T550, CAS No.: 18660-81-6 | 1 µM |
| Chemical compound, drug | 4-Aminopyridine | Sigma-Aldrich | Cat# A78403, CAS No.: 504-24-5 | 0.3–600 µM |
| Software, algorithm | pCLAMP 10.5 | Molecular Devices | RRID:SCR_014284 | |
| Software, algorithm | Axograph 1.7.2 | AxoGraph | RRID:SCR_014284 | |
| Software, algorithm | PRISM 7.0e | GraphPad Software | RRID:SCR_002798 | |
| Software, algorithm | ImageJ 1.5 | N.I.H. (USA) | RRID:SCR_003070 | |
| Software, algorithm | Adobe Photoshop CS6 | Adobe, USA | RRID:SCR_014199 | |
| Software, algorithm | R software 3.3.2 | Cran project (https://cran.r-project.org/) | RRID:SCR_001905 | |
| Software, algorithm | XPP-Aut 8.0 | University of Pittsburgh; Pennsylvania; USA | RRID:SCR_001996 | |

## Isolated SC preparation

Experiments were performed in accordance with European Community guiding principles on the care and use of animals (86/609/CEE, CE Off J no. L358, 18 December 1986), French decree no. 97/748 of 19 October 1987 (J Off République Française, 20 October 1987, pp. 12245–12248). All procedures were carried out in accordance with the local ethics committee of local universities and recommendations from the CNRS. We used Gad1$^{GFP}$ knock-in mice to visualize putative GABAergic INs (*Tamamaki et al., 2003*), as in our previous study (*Boeri et al., 2018*). To obtain E12.5–E16.5 Gad1$^{GFP}$ embryos, 8–12-week-old wild-type Swiss female mice were crossed with Gad1$^{GFP}$ Swiss male mice.

Isolated mouse SCs from 420 embryos were used in this work and obtained as previously described (*Delpy et al., 2008*; *Scain et al., 2010*). Briefly, pregnant mice were anesthetized by intramuscular injection of a mix of ketamine and xylazine and sacrificed using a lethal dose of $CO_2$ after embryos of either sex were removed. Whole SCs were isolated from eGFP-positive embryos and maintained in an artificial cerebrospinal fluid (ACSF) containing 135 mM NaCl, 25 mM NaHCO$_3$, 1 mM NaH$_2$PO$_4$, 3 mM KCl, 11 mM glucose, 2 mM CaCl$_2$, and 1 mM MgCl$_2$ (307 mOsm/kg H$_2$O), continuously bubbled with a 95% O$_2$-5% CO$_2$ gas mixture.

In the lumbar SC of Gad1$^{GFP}$ mouse embryos, eGFP neurons were detected using 488 nm UV light. They were localized in the ventrolateral marginal zone between the motor columns and the ventral funiculi (*Stam et al., 2012*). Embryonic V1$^R$ identity was confirmed by the expression of the forkhead transcription factor Foxd3 (*Boeri et al., 2018*).

## Whole-cell recordings and analysis

The isolated SC was placed in a recording chamber and was continuously perfused (2 ml/min) at room temperature (RT) (22–26°C) with oxygenated ACSF. Whole-cell patch-clamp recordings of lumbar spinal embryonic V1$^R$ were carried out under direct visualization using an infrared-sensitive CCD video camera. Whole-cell patch-clamp electrodes with a resistance of 4–7 MΩ were pulled from thick-wall borosilicate glass using a P-97 horizontal puller (Sutter Instrument Co., USA). They were filled with a solution containing (in mM): 96.4 K methanesulfonate, 33.6 KCl, 4 MgCl$_2$, 4 Na$_2$ATP, 0.3 Na$_3$GTP, 10 EGTA, and 10 HEPES (pH 7.2; 290 mOsm/kg-H$_2$O). This intracellular solution led to an equilibrium potential of chloride ions, $E_{Cl}$, of about –30 mV, close to the physiological values measured at E12.5 in spinal MNs (*Delpy et al., 2008*). The junction potential (6.6 mV) was systematically corrected offline.

Signals were recorded using Multiclamp 700B amplifiers (Molecular Devices, USA). Data were low-pass filtered (2 kHz), digitized (20 kHz) online using Digidata 1440A or 1550B interfaces, and acquired using pCLAMP 10.5 software (Molecular Devices, USA). Analyses were performed offline using pCLAMP 10.5 software packages (Molecular Devices; RRID:SCR_014284) and Axograph 1.7.2 (AxoGraph; RRID:SCR_002798).

In voltage-clamp mode, voltage-dependent K$^+$ currents ($I_{Kv}$) were elicited in the presence of 1 µM TTX (Alomone Labs, Cat# T550, CAS No.: 18660-81-6) by 500 ms depolarizing voltage steps (10 mV increments, 10 s interval) after a prepulse of 300 ms at $V_H$ = –100 mV. To isolate $I_{Kdr}$, voltage steps were applied after a 300 ms prepulse at $V_H$ = –30 mV that inactivated the low threshold transient potassium current $I_A$. $I_A$ was then obtained by subtracting offline $I_{Kdr}$ from the total potassium current $I_{Kv}$. Capacitance and leak current were subtracted using online P/4 protocol provided by pCLAMP 10.5.

In current-clamp mode, V1$^R$ discharge was elicited using 2 s depolarizing current steps (from 0 to ≈ 50 pA in 5–10 pA increments, depending on the input resistance of the cell) with an 8 s interval to ensure that the membrane potential returned to $V_H$. When a cell generated a sustained discharge, the intensity of the depolarizing pulse was reduced to the minimal value compatible with repetitive firing.

$I_{Nap}$ was measured in voltage-clamp mode using a 70 mV/s depolarizing voltage ramp (*Huang and Trussell, 2008*). This speed was slow enough to preclude substantial contamination by the inactivating transient current and fast enough to avoid substantial inactivation of $I_{Nap}$. Subtraction of the current evoked by the voltage ramp in the presence of 1 µM TTX from the control voltage ramp-evoked current revealed $I_{Nap}$.

## Pharmacological reagents

During patch-clamp recordings, bath application of TTX (1 µM, Alomone Labs, Cat# T550, CAS No.: 18660-81-6) or 4-AP (Sigma-Aldrich Cat# T550, CAS No.: 18660-81-6) was done using 0.5 mm diameter quartz tubing positioned, under direct visual control, 50 µm away from the recording area. The quartz tubing was connected to six solenoid valves linked with six reservoirs via a manifold. Solutions were gravity-fed into the quartz tubing. Their application was controlled using a VC-8 valve controller (Warner Instruments, USA).

4-AP was used to block $I_{Kdr}$. To determine the concentration–response curve, $I - V$ curves of $I_{Kdr}$ for different concentrations of 4-AP (0.3–300 µM) were compared to the control curve obtained in the absence of 4-AP. The percentage of inhibition for a given concentration was calculated by dividing the peak intensity of $I_{Kdr}$ by the peak value obtained in control condition. The obtained normalized concentration–response curves were fitted using the Hill equation:

$$\frac{100 - I_{min}}{1 + ([4-AP]/IC_{50})^{n_H}} + I_{min}$$

where [4-AP] is the 4-AP concentration, $I_{min}$ is the residual current (in percentage of the peak $I_{Kdr}$), $100 - I_{min}$ is the maximal inhibition achieved for saturating concentration of 4-AP, $IC_{50}$ is the 4-AP concentration producing half of the maximal inhibition, and $n_H$ is the Hill coefficient. Curve fitting was performed using KaleidaGraph 4.5 (Synergy Software, USA).

## Immunohistochemistry and confocal microscopy

E14.5 embryos were collected from pregnant females. Once dissected out of their yolk sac, SCs were dissected and immediately immersion-fixed in phosphate buffer (PB 0.1 M) containing 4% paraformaldehyde (PFA; freshly prepared in PB, pH 7.4) for 1 hr at 4°C. Whole SCs were then rinsed out in 0.12 M PB at 4°C, thawed at RT, washed in PBS, incubated in NH$_4$Cl (50 mM), diluted in PBS for 20 min, and then permeabilized for 30 min in a blocking solution (10% goat serum in PBS) with 0.2% Triton X-100. They were incubated for 48 hr at 4°C in the presence of the following primary antibodies: guinea pig anti-FoxD3 (1:5000, gift from Carmen Birchmeier and Thomas Müller of the Max Delbrück Center for Molecular Medicine in Berlin) and rabbit anti-cleaved Caspase-3 (1:1000, Cell Signaling Technology Cat# 9661, RRID:AB_2341188). SCs were then washed in PBS and incubated for 2 hr at RT with secondary fluorescent antibodies (goat anti-rabbit-conjugated 649; donkey anti-guinea pig-conjugated Alexa Fluor 405 [1:1000, ThermoFisher]) diluted in 0.2% Triton X-100 blocking solution. After washing in PBS, SCs were dried and mounted in Mowiol medium (Millipore, Molsheim, France). Preparations were then imaged using a Leica SP5 confocal microscope. Immunostaining was observed using a 40× oil-immersion objective with a numerical aperture of 1.25, as well as with a 63× oil-immersion objective with a numerical aperture of 1.32. Serial optical sections were obtained with a Z-step of 1 µm (40×) and 0.2–0.3 µm (63×). Images (1024 × 1024; 12-bit color scale) were stored using Leica software LAS-AF and analyzed using ImageJ 1.5 (N.I.H., USA, RRID:SCR_003070) and Adobe Photoshop CS6 (Adobe, USA, RRID:SCR_014199) software.

## Cluster analysis

To classify the firing patterns of embryonic V1$^R$, we performed a hierarchical cluster analysis on a population of 163 cells. Each cell was characterized by three quantitative measures of its firing pattern (see legend of *Figure 5*). After normalizing these quantities to zero mean and unit variance, we performed a hierarchical cluster analysis using the hclust function in R 3.3.2 software (Cran project; https://cran.r-project.org/; RRID:SCR_001905) that implements the complete linkage method. The intercluster distance was defined as the maximum Euclidean distance between the points of two clusters, and, at each step of the process, the two closest clusters were merged into a single one, thus constructing progressively a dendrogram. Clusters were then displayed in data space using the dendromat function in the R package 'squash' dedicated to color-based visualization of multivariate data. The best clustering was determined using the silhouette measure of clustering consistency (*Rousseeuw, 1987*). The silhouette of a data point, based on the comparison of its distance to other points in the same cluster and to points in the closest cluster, ranges from −1 to 1. A value near 1 indicates that the point is well assigned to its cluster, a value near 0 indicates that it is close to the decision boundary between two neighboring clusters, and negative values may indicate incorrect

assignment to the cluster. This allowed us to identify an optimal number k of clusters by maximizing the overall average silhouette over a range of possible values for k (*Rousseeuw, 1987*) using the silhouette function in the R package 'cluster'.

## Biophysical modeling

To understand the relationship between the voltage-dependent membrane conductances and the firing patterns of embryonic V1$^R$, we relied on a single-compartment conductance-based model that included the leak current, the transient and persistent components of the sodium current, $I_{Nat}$ and $I_{Nap}$, a delayed rectifier potassium current $I_{Kdr}$, and the inactivating potassium current $I_A$ revealed by experiments. Voltage evolution then followed the equation

$$C_{in}\frac{dV}{dt} = G_{in}(V_r - V) + G_{Nat}m^3h(E_{Na} - V) + G_{Nap}m_p^3s(V_{Na} - V) + G_{Kdr}n^3(E_K - V)$$
$$+ G_A m_A h_A(E_K - V) + I \tag{1}$$

where $C_{in}$ is the input capacitance; $G_{in}$ the input conductance; $G_{Nat}$, $G_{Nap}$, $G_{Kdr}$, and $G_A$ the maximal conductances of the aforementioned currents; $m, m_p$, $n$, and $m_A$ their activation variables; $h$ the inactivation variable of $I_{Nat}$; $s$ the slow inactivation variable of $I_{Nap}$; and $h_A$ the inactivation variable of $I_A$. $V_r$ is the baseline potential imposed by ad hoc current injection in current-clamp experiments; $E_{Na}$ and $E_K$ are the Nernst potentials of sodium and potassium ions, and $I$ the injected current. All gating variables satisfied equations of the form

$$\tau_x\frac{dx}{dt} = x_\infty(V) - x,$$

where the (in)activation curves were modeled by a sigmoid function of the form

$$x_\infty = \frac{1}{1 + \exp\left(-(V - V_x)/k_x\right)}$$

with $k_x$ being positive for activation and negative for inactivation. The time constant $\tau_x$ was voltage-independent except for the inactivation variables $h$ and $s$. The activation variable $m_A$ of $I_A$ was assumed to follow instantaneously voltage changes.

The effect of channel noise was investigated with a stochastic realization of the model, where channels kinetics were described by Markov-type models, assuming a unitary channel conductance of 10 pS for all channels.

## Choice of model parameters

Most model parameters were chosen on the basis of experimental measurements performed in the present study or already reported (*Boeri et al., 2018*). Parameters that could not be constrained from our experimental data were chosen from an experimentally realistic range of values. $V_r$ was set at –60 mV as in experiments (see *Table 1*). $C_{in}$ (average 13.15 pF, 50% between 11.9 and 15.1 pF, only 18 cells out of 246 in the first quartile below 7.2 pF or in the fourth quartile above 19 pF) and $G_{in}$ (50% of cells between 0.71 and 1.18 nS, only 7 out of 242 with input conductance above 2 nS) were not spread much in the cells recorded at E12.5, which showed that most embryonic V1$^R$ were of comparable size. Interestingly, $C_{in}$ and $G_{in}$ were not correlated, which indicated that the input conductance was determined by the density of leak channels rather than by the sheer size of the cell. Moreover, no correlation was observed between the passive properties and the firing pattern (*Boeri et al., 2018*). Therefore, we always set $G_{in}$ and $C_{in}$ to 1 nS and 13 pF in the model (except in *Figure 6—figure supplement 2*), close to the experimental medians (0.96 nS and 13.15 pF, respectively). The membrane time constant $C_{in}/G_{in}$ was then equal to 13 ms, which was also close to the experimental median (13.9 ms, N = 241).

$E_{Na}$ was set to 60 mV (see *Boeri et al., 2018*). The activation curve of $I_{Nap}$ was obtained by fitting experimental data, leading to an average mid-activation of –36 mV and an average steepness of 9.5 mV. The experimentally measured values of $G_{Nap}$ were in the range 0–2.2 nS. We assumed that the activation curve of $I_{Nat}$ was shifted rightward by 10 mV in comparison to $I_{Nap}$. No experimental data was available for the inactivation of $I_{Nat}$. We chose a mid-inactivation voltage $V_h$ = –45 mV and a steepness $k_h$ = –5 mV. We also assumed that the activation time constant of both $I_{Nat}$ and $I_{Nap}$ was

**Table 1.** Model parameters.

| Parameter | Basic model | Model with slow inactivation of I$_{Nap}$ |
|---|---|---|
| *Passive parameters* | | |
| Input conductance $G_{in}$ | 1 nS | Same |
| Input capacitance $C_{in}$ | 13 pF (E12.5, **Figures 7B, C, D, F** and **8B– D**) or 18 pF (E14.5, **Figure 7G**) | 13 pF |
| Resting potential $V_r$ | −60 mV | Same |
| Injected current I | 20 pA (**Figure 7B–G**) | 10 pA (**Figure 8C**) or 12 pA (**Figure 8D**) variable in **Figure 8B** |
| *Transient sodium current I$_{nat}$* | | |
| Maximal conductance $G_{Nat}$ | 20 nS | Same |
| Reversal potential $E_{Na}$ | 60 mV | |
| Activation exponent | 3 | |
| Mid-activation $V_m$ | −26 mV | |
| Steepness of activation $k_m$ | 9.5 mV | |
| Activation time constant | 1.5 ms | |
| Mid-inactivation $V_h$ | −45 mV | |
| Steepness of inactivation $K_h$ | −5 mV | |
| Inactivation time constant $\tau_m$ | Voltage-dependent (see Material s and methods) | |
| *Persistent sodium current I$_{Nap}$* | | |
| Maximal conductance | Variable (see text and figure captions) | Same |
| Mid-activation voltage | −36 mV | Same |
| Mid-inactivation $V_s$ | | −30 mV |
| Steepness of inactivation $k_s$ | | −5 mV |
| Inactivation time constant | Slow inactivation not included | 2 s |
| *Delayed rectifier potassium current I$_{Kdr}$* | | |
| Maximal conductance $G_{Kdr}$ | Variable (see text and figure captions) | Same |
| Reversal potential $E_K$ | −96 mV | |
| Activation exponent | 3 | |
| Mid-activation $V_n$ | −20 mV | |
| Steepness of activation $k_n$ | 15 mV | |
| Activation time constant $\tau_m$ | 10 ms | |
| *Potassium A current I$_A$ (when included in the basic model)* | | |
| Maximal conductance $G_A$ | Equal to $G_{Kdr}$ | Never included |
| Mid-activation $V_{mA}$ | −30 mV | |
| Steepness of activation $k_{mA}$ | 12 mV | |
| Activation time constant | Instantaneous activation | |
| Mid-inactivation $V_{hA}$ | −70 mV | |
| Steepness of inactivation $k_{hA}$ | −7 mV | |
| Inactivation time constant $\tau_{hA}$ | 23 ms | |

1.5 ms, and that the inactivation time constant was voltage-dependent: $\tau_h(V) = 16.5 - 13.5 \tanh((V + 20)/15)$, decreasing by one order of magnitude (from 30 ms down to 3 ms) with the voltage increase. This enabled us to account for the shape of the APs recorded in experiments, showing a slow rise time and rather long duration. The conductance $G_{Nat}$ was not measured experimentally. When choosing a reasonable value of 20 nS for $G_{Nat}$, the model behaved very much as recorded embryonic V1$^R$: with similar current threshold (typically 10–20 pA) and stable plateau potentials obtained for the largest values of $G_{Nap}$.

When taking into account slow inactivation of $I_{Nap}$ (see *Figure 8*), we chose $V_s = –30$ mV for the mid-inactivation voltage and set the steepness $k_s$ at –5 mV (as for the inactivation of $I_{Nat}$). For simplicity, we assumed that the inactivation time constant was voltage-independent and set at 2 s.

$E_K$ was set to the experimentally measured value of –96 mV (*Boeri et al., 2018*). The activation parameters of $I_{Kdr}$ were obtained by fitting the experimental data: $V_n = –20$ mV, $k_n = 15$ mV, $\tau_n = 10$ ms, and an activation exponent of 3. The activation and inactivation properties of $I_A$ were also chosen based on experimental measurements. Accordingly, $V_{m_A} = –30$ mV, $k_{m_A} = –12$ mV, $V_{h_A} = –70$ mV, $k_h A = –7$ mV, and $\tau_{h_A} = 23$ ms. When $I_A$ was taken into account, we assumed that $G_A = G_{Kdr}$, consistently with experimental data (see *Figure 6—figure supplement 1*).

## Numerical simulations and dynamical systems analysis

We integrated numerically the deterministic model using the freeware XPPAUT 8.0 (University of Pittsburgh; Pennsylvania; USA; RRID:SCR_001996) (*Ermentrout, 2002*) and a standard fourth-order Runge–Kutta algorithm. XPPAUT was also used to compute one-parameter and two-parameter bifurcation diagrams. The stochastic version of the model was also implemented in XPPAUT and computed with a Gillespie's algorithm (*Gillespie, 1976*).

To investigate the dynamics of the model with slow inactivation of $I_{Nap}$, we relied on numerical simulations together with fast/slow dynamics analysis (*Witelski and Bowen, 2015*). In this approach, one distinguishes slow dynamical variables (here only *s*) and fast dynamical variables. Slow variables vary little at the time scale of fast variables and may therefore be considered as constant parameters of the fast dynamics in first approximation. In contrast, slow variables are essentially sensitive to the time average of the fast variables, much more than to their instantaneous values. This separation of time scales allows one to conduct a phase plane analysis of the full dynamics.

## Statistics

Samples sizes (n) were determined based on previous experience. The number of embryos (N) is indicated in the main text and figure captions. No power analysis was employed, but sample sizes are comparable to those typically used in the field. All values were expressed as mean with standard error of mean (SEM). Statistical significance was assessed by non-parametric Kruskal–Wallis test with Dunn's post hoc test for multiple comparisons, Mann–Whitney test for unpaired data or Wilcoxon matched pairs test for paired data using GraphPad Prism 7.0e Software (USA). Significant changes in the proportions of firing patterns with age were assessed by chi-square test for large sample and by Fisher's exact test for small sample using GraphPad Prism 7.0e Software (GraphPad Software; RRID: SCR_002798). Significance was determined as *p<0.05, **p<0.01, or ***p<0.001. The exact p values are mentioned in the result section or in the figure captions.

## Acknowledgements

We thank Susanne Bolte, Jean-François Gilles, and France Lam for assistance with confocal imaging (IBPS imaging facility) and IBPS rodent facility team for animal care and production. We thank University Paris Descartes for hosting Yulia Timofeeva as an invited professor. This work was supported by INSERM, CNRS, Sorbonne Université (Paris), Université de Bordeaux, Université Paris Descartes, and Fondation pour la Recherche Médicale.

## Additional information

### Funding

| Funder | Grant reference number | Author |
|---|---|---|
| Fondation pour la Recherche Médicale | DEQ20160334891 | Pascal Legendre |

The funders had no role in study design, data collection and interpretation, or the decision to submit the work for publication.

### Author contributions

Juliette Boeri, Formal analysis, Investigation; Claude Meunier, Conceptualization, Resources, Software, Formal analysis, Supervision, Validation, Investigation, Visualization, Methodology, Writing - review and editing; Hervé Le Corronc, Formal analysis, Validation, Investigation, Visualization, Methodology; Pascal Branchereau, Formal analysis, Supervision, Validation, Investigation, Visualization, Methodology, Writing - review and editing; Yulia Timofeeva, Conceptualization, Resources, Software, Formal analysis, Validation, Investigation, Visualization, Methodology, Writing - review and editing; François-Xavier Lejeune, Data curation, Software, Formal analysis, Validation, Investigation, Methodology; Christine Mouffle, Resources, Data curation; Hervé Arulkandarajah, Investigation, Methodology; Jean Marie Mangin, Writing - review and editing; Pascal Legendre, Conceptualization, Resources, Data curation, Software, Supervision, Funding acquisition, Validation, Visualization, Methodology, Writing - original draft, Project administration, Writing - review and editing; Antonny Czarnecki, Conceptualization, Data curation, Formal analysis, Supervision, Validation, Investigation, Methodology, Writing - original draft

### Author ORCIDs

Claude Meunier (iD) https://orcid.org/0000-0002-8216-3991
Pascal Branchereau (iD) https://orcid.org/0000-0003-3972-8229
Yulia Timofeeva (iD) https://orcid.org/0000-0003-3178-7830
Pascal Legendre (iD) https://orcid.org/0000-0002-5086-4515
Antonny Czarnecki (iD) https://orcid.org/0000-0002-5104-034X

### Ethics

Animal experimentation: Experiments were performed in accordance with European Community guiding principles on the care and use of animals (86/609/CEE, CE Off J no. L358, 18 December 1986), French decree no. 97/748 of October 19, 1987 (Journal Officiel République Française, 20 October 1987, pp. 12245-12248). All procedures were carried out in accordance with the local ethics committee of local Universities and recommendations from the CNRS. pregnant mice were anesthetized by intramuscular injection of a mix of ketamine and xylazine and sacrificed using a lethal dose of $CO_2$ after embryos of either sex were removed. Every effort was made to minimize suffering.

### Decision letter and Author response

Decision letter https://doi.org/10.7554/eLife.62639.sa1
Author response https://doi.org/10.7554/eLife.62639.sa2

## Additional files

### Supplementary files

• Transparent reporting form

### Data availability

All data generated or analysed during this study are included in the manuscript and supporting files. Source data files have been provided for Figures 3 (Source data files for cluster analysis).

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
