## [Decision Letter]

**Acceptance summary:**

Boeri and colleagues studied the developmental emergence and transformations of electrical activity patterns of embryonic mouse Renshaw neurons in the spinal cord by a novel combination of rigorous electrophysiological and biophysical modeling analyses. Their studies indicate that a dynamic interaction of two prominently expressed sodium and potassium currents in these neurons can produce a variety of electrophysiological activity patterns and account for their transformations during embryonic development. These studies contribute importantly to understanding biophysical mechanisms by which spinal neurons express complex electrophysiological activity patterns including spontaneous activity during embryonic development.

**Decision letter after peer review:**

[Editors’ note: the authors submitted for reconsideration following the decision after peer review. What follows is the decision letter after the first round of review.]

Thank you for submitting your work entitled "Two voltage-dependent currents can explain the functional diversity of embryonic Renshaw cells" for consideration by *eLife*. Your article has been reviewed by 3 peer reviewers, including Jeffrey C Smith as the Reviewing Editor and Reviewer #1, and the evaluation has been overseen by a Senior Editor. The following individual involved in review of your submission has agreed to reveal their identity: Ryan S Phillips (Reviewer #2).

Our decision has been reached after consultation between the reviewers. Based on these discussions and the individual reviews below, we regret to inform you that your work will not be considered further for publication in *eLife*.

The reviewers generally thought that your combination of electrophysiological and modeling analyses has the potential to explain the patterns of Renshaw neuron activity in terms of biophysical properties that you characterize. There was also agreement that your analyses provide a more detailed and potentially valuable developmental analysis of neuronal activity patterns than done previously for any mouse spinal neuron type during the developmental window studied. However, after discussion, the reviewers agreed that the present studies have not yet made the advance of convincingly connecting the developmental patterns of Renshaw neuron activity to the developmental patterns of spontaneous neural activity in the spinal cord, which is how the authors are trying to frame the paper.

*Reviewer #1:*

This very well written manuscript presents an extensive set of experimental observations with rigorous electrophysiological and sophisticated modeling analyses of how dynamically interacting sodium and potassium currents can produce different neuronal firing patterns in Renshaw cells (V1R) during mouse embryonic spinal cord development. The authors analyze firing patterns of V1R during the important developmental period when spontaneous neural activity (SNA) occurs in the mouse embryonic spinal cord (E11.5-E14.5), and during the critical period (E14.5-E16.5) when GABAergic neurotransmission shifts from excitation to inhibition and rhythmic locomotor-like activity emerges. The important finding is that there appear to be five distinct functional classes of V1R transiently present at the onset of SNA, and this functional diversity shifts as development proceeds to the critical period. The authors present substantial evidence from their experimental electrophysiological/pharmacological and biophysical modeling analyses that their observed diversity of firing patterns and the developmental transformations can be attributed largely to the dynamical synergy between two important voltage-dependent currents- the delayed rectifier potassium current and a persistent, TTX-sensitive sodium current- that the authors document from their electrophysiological measurements and can explain from their modeling analyses. These analyses provide a much more detailed view than previous ideas about patterns of emerging activity at a neuron level during embryonic spinal cord development.

1. The authors conclude that a "single mechanism" involving two voltage-gated channels with opposite functions that are ubiquitous in neurons can produce functional diversity between neurons. This is a broad statement that may pertain to the Renshaw cells studied to explain their activity patterns, but it is not at all certain that this explains activity patterns of other spinal cord neurons during development. This conclusion needs to be tempered. The authors do a nice job trying to sort out the potential contributions of IA, IKdr, INaP, and leak currents at experimental and theoretical levels. But there is no mention of calcium currents, for example, that are undoubtedly in the mix developmentally. Mixed-cationic conductances may also be involved. It is not clear that the results of these analyses apply to "many classes of embryonic cells in mammals at an early stage of development" (p. 24, line 560) without studying other embryonic cells.

2. Related to point 1, the description of experimental procedures for the voltage-clamp electrophysiological analyses of contributions of potassium currents seems incomplete. Typically, in voltage-clamp recording analyses to isolate potassium currents, sodium and calcium currents should be blocked pharmacologically. It is not clear from the Methods description that this was the case. This requires further explanation.

3. Also related to point 1, the authors don't offer any ideas about how the transformations in neuronal electrophysiological behavior might be connected to the emergence of rhythmic activity at the endpoint of their developmental analysis even though they place their analysis in this context. There are other spinal interneuron populations involved in the emergence of such activity that may show a different developmental trajectory, such as the maintenance of a relatively high GNaP/GKdr ratio and also involvement of GKleak. This requires some discussion. It is also not clear that the authors provide a reasonable explanation for the results presented in Figure 12 where they attempt to connect the patterns of V1R activity to motoneuron activity to explain SNA. These experiments add more data to an already complex data set and analyses that can probably be eliminated.

*Reviewer #2:*

In this manuscript, Boeri et al., investigate how the balance of a persistent sodium current (INaP) and a delayed rectifier potassium current (IKdr) shape the diversity of firing patterns in Renshaw cells in embryonic development. This study is a good example of the synergistic use of computational and experimental approaches. The authors identify four distinct types of activity patterns and demonstrate that pharmacological blockade of IKdr transforms firing patterns in a predictable sequence. Furthermore, they find differences in the maximal conductances of INaP and Ikdr which suggest that firing patterns may be determined by the ratio of these two conductances. Finally, they use computational simulations to demonstrate how the balance of INaP and Ikdr can explain the diversity of firing patterns in a model neuron. Overall I am enthusiastic about this work but have some concerns.

1. In Figure 11 C & D it is not clear what the trajectories in the GNaP-Vm plane represent since GNaP is a parameter not a dynamical variable. Does the trajectory represent GNaP*m^3*s? In 11D, is GNaP = 2.5nS as in D?

2. The basic model does a great job capturing and explaining how the GNaP/GKdr can determine the firing firing pattern Figure 10. However, I have some questions about the robustness of the modeling predictions:

i. The RS region in the model is in very good agreement with the experimental data. How sensitive is this fit to changes in INaP and IKdr activation dynamics? For example how would using the INaP activation dynamics reported in Boeri et al., 2018 change the RS region?

ii. For the simulations in Figure 10 the model does not incorporate any slow inactivation of INaP. If inactivation was included would the location of the RS region shift to the right in the GNaP-GKdr plane since a larger GNaP would be required to generate the same current?

iii. In the RS region in Figure 10 it is likely sensitive to the strength of the applied current. In the experimental data the applied current appears to range from 15-50 pA however in the model the RS region is predicted using an applied current of 20pA. Is the fit between the predicted RS region and the data as good with an applied current of 15pA or 50pA?

3. In order to explain the sequential change in V1R activity patterns with progressive block of IKdr the model requires the proportional reduction of Gin and the applied current. The applied current was not varied in any of the example traces presented in Figure 4. Why was it required in the model? Can the model still explain these transitions without reducing the applied current?

4. The explanation of the repetitive plateaus requires inactivation of INaP for the switch from the plateau to the quiescent state. In the model this results in a relatively strong slope during the plateau state and a relatively gradual transition to the quiescent state compared to the example shown in Figure 11A. Inactivation of INaP is still a reasonable explanation, however other burst termination mechanisms could explain these transitions and should be discussed. Also, does the model suggest that only neurons with repetitive plateaus and mixed events have inactivating INaP?

*Reviewer #3:*

This paper addresses important issues about biophysical mechanisms involved in the generation of spontaneous network activity in the developing spinal cord. Pharmacological and electrophysiological analysis are performed to characterize membrane properties of Renshaw cells during embryonic development in the mouse. The authors demonstrate the existence of heterogeneous firing properties relying on the balance between two opposing voltage-dependent conductances, the persistent sodium current (INaP) and the delayed rectifier potassium current (IKdr). A clear description is provided about how authors classified Renshaw neurons into 4 groups (long-lasting plateau potentials, mixture of spikes and short lasting bursts, repetitive spiking and single spiking) based on biophysical properties. Using both experiments and modeling, the authors show that the balance between INaP and IKdr in Renshaw neurons accounts for functional differences during development. Specifically, cells expressing bistable behaviors have the higher INaP/IKdr ratio, while single spiking cells have the lower INaP/IKdr ratio. Also, an unexpected developmental change in the firing pattern of Renshaw cells is described that switch from repetitive spiking or plateau potential patterns at E11.5-E12.5 to a dominant single-spiking pattern at E13.5-E16.5. The authors suggest that the above-mentioned change may be due to a developmental increase in IKdr. In line with this, when IKdr is decreased by 4-AP most of single spiking neurons recorded at E14.5 switch to an INaP-mediated plateau potential state.

To tackle the physiological meaning of this developmental transition in the firing pattern of Renshaw cells, the authors recorded GABAergic inputs on motoneurons and bath-applied 4-AP in isolated spinal cords at E12.5. The 4-AP-induced increase of GABAergic inputs evoked by a cervical stimulation was attributed to an increase in the excitability of Renshaw cells by favoring the emergence of repetitive firing and plateau potentials. However, we do not have direct evidence of it. These data appear to be over-interpreted insofar as IKdr is not specific to Renshaw cells. In particular, IKdr is also expressed in motoneurons and may thus influence their excitability. Furthermore, the approach of using cervical stimulation to induce GABAergic inputs onto motoneurons rather than recording spontaneous activities is surprising in the context of this study.

Overall, the authors convincingly state that INaP interacts with the IKdr to regulate the firing patterns of Renshaw cells. However, the finding of a balance between inward and outward currents in governing the firing pattern of neurons is not novel. I am afraid that the biological insights afforded by the study on the biophysical mechanisms involved in the generation of spontaneous activities are not strong enough. My opinion is that the work does not make important breakthrough such that deserving to be published in *eLife*.

[Editors’ note: further revisions were suggested prior to acceptance, as described below.]

Thank you for choosing to send your work entitled "Two voltage-dependent currents can explain the functional diversity of embryonic Renshaw cells" for consideration at *eLife*. Your letter of appeal has been considered by a Senior Editor, and the Reviewing editor in consultation with previous Reviewers, and we are prepared to consider a revised submission incorporating the changes indicated in your letter of appeal with no guarantees of acceptance.

To assist you in preparing your revised submission, we are communicating the following assessment in response to your appeal letter by one of the previous reviewers, who raised important concerns to be addressed in addition to your other proposed revisions.

Essential revisions:

The authors carefully consider most of my concerns. They raise a disagreement with my major concern about the lack of novelty of the main conclusion of the paper, stipulating that a simple mechanism involving two opposite slowly inactivating voltage-gated channels is sufficient to produce functional diversity in neurons. This conclusion appears to me very close to that of previous papers (see references below) where combined experimental and modeling studies show how two opposing currents shape diversity of the firing patterns (silent, spiking, bursting) in a population of neurons. None of these important studies in the field were cited. It would be interesting that the authors discuss these papers in respect to their own data and show how their main conclusion is different, deserving to be published in *eLife*.

1. Contribution of persistent Na^+^ current and M-type K^+^ current to somatic bursting in CA1 pyramidal cells: combined experimental and modeling study. David Golomb 1, Cuiyong Yue, Yoel Yaari J Neurophysiol. 2006 Oct;96(4):1912-26. doi: 10.1152/jn.00205.2006. Epub 2006 Jun 28.

2. Competition between Persistent Na + and Muscarine-Sensitive K + Currents Shapes Perithreshold Resonance and Spike Tuning in CA1 Pyramidal Neurons. Jorge Vera 1, Julio Alcayaga 1, Magdalena Sanhueza. Front Cell Neurosci. 2017 Mar 8;11:61. doi: 10.3389/fncel.2017.00061.

3. Intrinsic bursting activity in the pre-Bötzinger complex: role of persistent sodium and potassium currents. Ilya A Rybak 1, Natalia A Shevtsova, Krzysztof Ptak, Donald R McCrimmon. Biol Cybern 2004 Jan;90(1):59-74. doi: 10.1007/s00422-003-0447-1. Epub 2004 Jan 21.

4. Persistent Sodium Current, Membrane Properties and Bursting Behavior of Pre-Bötzinger Complex Inspiratory Neurons in vitro Christopher A. Del Negro, Naohiro Koshiya*, Robert J. Butera Jr. and Jeffrey C. Smith 01 NOV 2002, https://doi.org/10.1152/jn.00081.2002.

---

## [Author Response]

[Editors’ note: The authors appealed the original decision. What follows is the authors’ response to the first round of review.]

Reviewer #1:This very well written manuscript presents an extensive set of experimental observations with rigorous electrophysiological and sophisticated modeling analyses of how dynamically interacting sodium and potassium currents can produce different neuronal firing patterns in Renshaw cells (V1R) during mouse embryonic spinal cord development. The authors analyze firing patterns of V1R during the important developmental period when spontaneous neural activity (SNA) occurs in the mouse embryonic spinal cord (E11.5-E14.5), and during the critical period (E14.5-E16.5) when GABAergic neurotransmission shifts from excitation to inhibition and rhythmic locomotor-like activity emerges. The important finding is that there appear to be five distinct functional classes of V1R transiently present at the onset of SNA, and this functional diversity shifts as development proceeds to the critical period. The authors present substantial evidence from their experimental electrophysiological/pharmacological and biophysical modeling analyses that their observed diversity of firing patterns and the developmental transformations can be attributed largely to the dynamical synergy between two important voltage-dependent currents- the delayed rectifier potassium current and a persistent, TTX-sensitive sodium current- that the authors document from their electrophysiological measurements and can explain from their modeling analyses. These analyses provide a much more detailed view than previous ideas about patterns of emerging activity at a neuron level during embryonic spinal cord development.1. The authors conclude that a "single mechanism" involving two voltage-gated channels with opposite functions that are ubiquitous in neurons can produce functional diversity between neurons. This is a broad statement that may pertain to the Renshaw cells studied to explain their activity patterns, but it is not at all certain that this explains activity patterns of other spinal cord neurons during development. This conclusion needs to be tempered.

Obviously our intention was not to claim that this was the only mechanism leading to a diversity of activity patterns in neurons. Our intention was to elucidate from an experimental and theoretical points of view how the expression of a limited repertoire of voltage-gated channels can lead, as a core mechanism, to functional diversity in immature, which was not demonstrated in biological conditions yet. This point was corrected both in the introduction (page 3 lines 90-92) and in the discussion (pages 2122 lines 517-531) of the new version of our manuscript.

It is well known that in the adult the expression of several voltage-gated channels subtypes is required for functional heterogeneity between neuronal subtypes and to stabilize a particular firing pattern in a neuronal population, as observed, for instance, in mature Renshaw cells. Contrarily to the mature CNS (O'Leary T et al., 2013), there is no homeostatic regulation of channel density to ensure the robustness of V1^R^ excitability during their early development. In addition, the limited repertoire of voltage-gated channels at this developmental stage precludes channel degeneracy, which is also known to ensure the robustness of excitability in mature neurons (O'Leary T et al., 2013). The discussion was rewritten accordingly (pages 23-24 556-574).

The authors do a nice job trying to sort out the potential contributions of IA, IKdr, INaP, and leak currents at experimental and theoretical levels. But there is no mention of calcium currents, for example, that are undoubtedly in the mix developmentally. Mixed-cationic conductances may also be involved.

We did not observe any calcium current during voltage-clamp experiments in V1^R^ before E14.5 (This is now mentioned on page 5 lines 132-136). But we agree with the reviewer that the addition of other non-inactivating or slowly inactivating inward currents, such as I_CAN_, could also lead to functional diversity. This is now mentioned on page 22 lines 528-531.

It is not clear that the results of these analyses apply to "many classes of embryonic cells in mammals at an early stage of development" (p. 24, line 560) without studying other embryonic cells.

We discarded this sentence that was misleading. We did not mean to say that functional diversity occurs in all developing neurons. What our experimental and our computational analyses show is that a simple mechanism involving only two voltagegated channels with opposite effects is sufficient to produce functional diversity in immature neurons having a limited repertoire of voltage-gated channels. The discussion was modified accordingly (page 23 lines 563-567).

2. Related to point 1, the description of experimental procedures for the voltage-clamp electrophysiological analyses of contributions of potassium currents seems incomplete. Typically, in voltage-clamp recording analyses to isolate potassium currents, sodium and calcium currents should be blocked pharmacologically. It is not clear from the Methods description that this was the case. This requires further explanation.

We apologize for omitting this information. All experiments were performed in the presence of TTX to block sodium currents (see page 4 lines 107-112 and page 26 lines 616-617). This now corrected in Materials and methods section. We did not observe any calcium current at this developmental stage in Renshaw cells (Boeri et al., 2018). Additional experiments were performed to compare the effect of external calcium removal on the I-V relationship of potassium current. We did not observe any change in the I-V curves of I_A_ or I_KdR_ when external calcium was removed, indicating that calciumdependent potassium currents were not yet present in E12.5 V1^R^. This is now mentioned in the result section page 4 lines 115-116 and page 10 lines 253-255.

3. Also related to point 1, the authors don't offer any ideas about how the transformations in neuronal electrophysiological behavior might be connected to the emergence of rhythmic activity at the endpoint of their developmental analysis even though they place their analysis in this context. There are other spinal interneuron populations involved in the emergence of such activity that may show a different developmental trajectory, such as the maintenance of a relatively high GNaP/GKdr ratio and also involvement of GKleak. This requires some discussion. It is also not clear that the authors provide a reasonable explanation for the results presented in Figure 12 where they attempt to connect the patterns of V1R activity to motoneuron activity to explain SNA. These experiments add more data to an already complex data set and analyses that can probably be eliminated.

We agree with the reviewer that this part of the Results section is beyond the scope of the paper and can be misleading. In accordance with the reviewer’s suggestion we decided to discard these results, as they did not add any pertinent information according to the scope of the paper. The Materials and methods section and the discussion were modified accordingly.

Reviewer #2:In this manuscript, Boeri et al., investigate how the balance of a persistent sodium current (INaP) and a delayed rectifier potassium current (IKdr) shape the diversity of firing patterns in Renshaw cells in embryonic development. This study is a good example of the synergistic use of computational and experimental approaches. The authors identify four distinct types of activity patterns and demonstrate that pharmacological blockade of IKdr transforms firing patterns in a predictable sequence. Furthermore, they find differences in the maximal conductances of INaP and Ikdr which suggest that firing patterns may be determined by the ratio of these two conductances. Finally, they use computational simulations to demonstrate how the balance of INaP and Ikdr can explain the diversity of firing patterns in a model neuron. Overall I am enthusiastic about this work but have some concerns.

We greatly appreciate the opinion of the reviewer on our work. We answered all reviewer’s remarks and further simplified the theoretical analysis to make it more accessible to the reader. In particular, we no longer mention Bautin bifurcations. We also modified Figure 8 to make the explanation of the bursting scenarios clearer.

1. In Figure 11 C & D it is not clear what the trajectories in the GNaP-Vm plane represent since GNaP is a parameter not a dynamical variable. Does the trajectory represent GNaP*m^3*s? In 11D, is GNaP = 2.5nS as in D?

Figures 11C and D (now modified figures 8C and D) show the bifurcation diagrams of the basic model (without slow inactivation) when G_Nap_ is the main parameter. We then took into account the slow inactivation (G_Nap_*mp^3*s) of I_Nap_, and the traces (now voltage V(t) and G_Nap_*s(t)) are shown on the right. We finally superimposed the voltage traces of the model with slow inactivation to the bifurcation diagrams to demonstrate how different phases of voltage dynamics in the presence of slow inactivation (G_Nap_*s(t)) are linked with the bifurcation diagrams of the basic model (where G_Nap_ is a constant parameter).

2. The basic model does a great job capturing and explaining how the GNaP/GKdr can determine the firing firing pattern Figure 10. However, I have some questions about the robustness of the modeling predictions:i. The RS region in the model is in very good agreement with the experimental data. How sensitive is this fit to changes in INaP and IKdr activation dynamics? For example how would using the INaP activation dynamics reported in Boeri et al., 2018 change the RS region?

The different regions, in particular the RS region, are indeed shifted when the parameters of the model are modified (downward shift when it is easier to activate I_Kdr_, leftward shift when it becomes easier to activate I_Nap_). However, the bifurcation diagram of the model remains qualitatively the same with SS, RS and PP regions. The parameters we chose are actually the same as in Boeri et al. 2018 (see explanation below).

ii. For the simulations in Figure 10 the model does not incorporate any slow inactivation of INaP. If inactivation was included would the location of the RS region shift to the right in the GNaP-GKdr plane since a larger GNaP would be required to generate the same current?

One indeed expects the diagram to move in that direction. However, a straightforward bifurcation analysis cannot be performed in this case. That is why we relied on the fact that inactivation of I_Nap_ is slow and performed a slow/fast analysis of the dynamics that enabled us to explain the bursting patterns of Renshaw cells. This is the standard approach to deal with these issues when studying the dynamics of neurons.

iii. In the RS region in Figure 10 it is likely sensitive to the strength of the applied current. In the experimental data the applied current appears to range from 15-50 pA however in the model the RS region is predicted using an applied current of 20pA. Is the fit between the predicted RS region and the data as good with an applied current of 15pA or 50pA?

The RS region is indeed shifted when the applied current is modified (see, for instance, new Figure 7—figure supplement 2, where both the applied current and the input conductance are modified), and the fit with data points can become less striking. However, the diagram is not qualitatively modified, which shows the robustness of the model, and the agreement with experimental data remains good. One must also keep in mind that a model with a fixed injected current (and fixed input conductance) cannot accurately account for the data, which were obtained for an injected current that varied from cell to cell and for cells with different intrinsic properties. In this respect, the agreement between the model and experimental points shown in new Figure 7F may be slightly misleading; we mention it now in the text (see pages 14-15 lines 359-365).

3. In order to explain the sequential change in V1R activity patterns with progressive block of IKdr the model requires the proportional reduction of Gin and the applied current. The applied current was not varied in any of the example traces presented in Figure 4. Why was it required in the model? Can the model still explain these transitions without reducing the applied current?

Decreasing only the input conductance shifts the whole bifurcation diagram to the left, but it also distorts the diagram and extends the RS region, making it indeed difficult to account for experimental results on the effect of 4-AP. In contrast, changing both the input conductance and the injected current shifts the RS region leftward and downward, as shown in Figure 7—figure supplement 2, and accounts for the experimental results. Changing both parameters in the model makes sense because 4-AP reduces the input conductance of the recorded cells. This decreases the rheobase current, and cells with smaller input conductance were actually recorded with a smaller injected current in experiments.

4. The explanation of the repetitive plateaus requires inactivation of INaP for the switch from the plateau to the quiescent state. In the model this results in a relatively strong slope during the plateau state and a relatively gradual transition to the quiescent state compared to the example shown in Figure 11A. Inactivation of INaP is still a reasonable explanation, however other burst termination mechanisms could explain these transitions and should be discussed. Also, does the model suggest that only neurons with repetitive plateaus and mixed events have inactivating INaP?

We checked that channel noise *per se* was not sufficient to elicit frequent transitions between states in the model in the absence of slow inactivation of I_Nap_. The only option, which is the usual explanation for bursting, is to incorporate a slow dynamical process. Both I_Nap_ and I_Kdr_ shows slow inactivation, but the inactivation of I_Kdr_ is much too slow to account for the observed bursting patterns. The details of the bursting patterns (slope of the plateau, abruptness of transitions, duration of episodes) depend on the model parameters (and on the cell recorded in experiments), but slow inactivation of I_Nap_ is sufficient to account qualitatively for both pseudo-plateau bursting and elliptic bursting. Slow inactivation of I_Nap_ induces bursting for cells displaying bistability between SS and PP regimes (pseudo-plateau bursting) or RS and PP regimes (elliptic bursting accounting for mixed events). It affects the repetitive discharge of cells that are well inside the RS domain, but does not induce bursting then. To summarize, all cells display slow inactivation of I_Nap_, which has an impact on the discharge, but only cells located near transitions between different firing patterns display bursting as a result of slow inactivation.

Reviewer #3:This paper addresses important issues about biophysical mechanisms involved in the generation of spontaneous network activity in the developing spinal cord. Pharmacological and electrophysiological analysis are performed to characterize membrane properties of Renshaw cells during embryonic development in the mouse. The authors demonstrate the existence of heterogeneous firing properties relying on the balance between two opposing voltage-dependent conductances, the persistent sodium current (INaP) and the delayed rectifier potassium current (IKdr). A clear description is provided about how authors classified Renshaw neurons into 4 groups (long-lasting plateau potentials, mixture of spikes and short lasting bursts, repetitive spiking and single spiking) based on biophysical properties. Using both experiments and modeling, the authors show that the balance between INaP and IKdr in Renshaw neurons accounts for functional differences during development. Specifically, cells expressing bistable behaviors have the higher INaP/IKdr ratio, while single spiking cells have the lower INaP/IKdr ratio. Also, an unexpected developmental change in the firing pattern of Renshaw cells is described that switch from repetitive spiking or plateau potential patterns at E11.5-E12.5 to a dominant single-spiking pattern at E13.5-E16.5. The authors suggest that the above-mentioned change may be due to a developmental increase in IKdr. In line with this, when IKdr is decreased by 4-AP most of single spiking neurons recorded at E14.5 switch to an INaP-mediated plateau potential state.To tackle the physiological meaning of this developmental transition in the firing pattern of Renshaw cells, the authors recorded GABAergic inputs on motoneurons and bath-applied 4-AP in isolated spinal cords at E12.5. The 4-AP-induced increase of GABAergic inputs evoked by a cervical stimulation was attributed to an increase in the excitability of Renshaw cells by favoring the emergence of repetitive firing and plateau potentials. However, we do not have direct evidence of it. These data appear to be over-interpreted insofar as IKdr is not specific to Renshaw cells. In particular, IKdr is also expressed in motoneurons and may thus influence their excitability. Furthermore, the approach of using cervical stimulation to induce GABAergic inputs onto motoneurons rather than recording spontaneous activities is surprising in the context of this study.

We agree that this type of experiment was misleading according to the scope of the paper. As stated by reviewer 1 our paper provides a much more detailed view than previous ideas about patterns of emerging activity at a neuron level during embryonic spinal cord development. Following the suggestion of reviewer 1 we removed this data (see above). The result section, the discussion and the Materials and methods section were modified accordingly.

Overall, the authors convincingly state that INaP interacts with the IKdr to regulate the firing patterns of Renshaw cells. However, the finding of a balance between inward and outward currents in governing the firing pattern of neurons is not novel. I am afraid that the biological insights afforded by the study on the biophysical mechanisms involved in the generation of spontaneous activities are not strong enough. My opinion is that the work does not make important breakthrough such that deserving to be published in eLife.

We do not agree with this assessment of the reviewer. Saying that our work confirms that the balance between inward and outward currents govern the firing pattern of neuron is spurious, as it does not take into account the theoretical and physiological scope of our work.

Indeed it is known from Hodgkin and Huxley works that the balance between delayed rectifier potassium currents and transient inward current underlies action potentials. But the diversity in the firing patterns classically observed in mature neurons was thought to result from a complex interplay between several classes of different voltage-gated channels (see Eve Marder’s work) being required to stabilize their firing pattern. Unless this reviewer can provide references clearly indicating that what we show is not novel at all we insist that functional diversity patterns within a neuron population controlled by the balance between two opposite currents only (namely a delay inward potassium current and a persistent inward current) is novel. Of course this is feasible in neurons in which the expression of several different classes of voltage-gated channel is limited as observed in immature neurons.

Our work clearly indicates that the balance between two slowly inactivating current of opposite direction only suffice to explain functional diversity between immature neurons of the same class. Our computational analysis also predicts the different limits within such a simple synergy in which a particular pattern can occur. It also explains why adding an additional slowly inactivating current can change the firing pattern of a developing neurons as previously observed in the brain stem (Chevalier et al. 2016).

[Editors’ note: what follows is the authors’ response to the second round of review.]

Essential revisionsThe authors carefully consider most of my concerns. They raise a disagreement with my major concern about the lack of novelty of the main conclusion of the paper, stipulating that a simple mechanism involving two opposite slowly inactivating voltage-gated channels is sufficient to produce functional diversity in neurons. This conclusion appears to me very close to that of previous papers (see references below) where combined experimental and modeling studies show how two opposing currents shape diversity of the firing patterns (silent, spiking, bursting) in a population of neurons. None of these important studies in the field were cited. It would be interesting that the authors discuss these papers in respect to their own data and show how their main conclusion is different, deserving to be published in eLife.1. Contribution of persistent Na^+^ current and M-type K^+^ current to somatic bursting in CA1 pyramidal cells: combined experimental and modeling study. David Golomb 1, Cuiyong Yue, Yoel Yaari J Neurophysiol. 2006 Oct;96(4):1912-26. doi: 10.1152/jn.00205.2006. Epub 2006 Jun 28.2. Competition between Persistent Na + and Muscarine-Sensitive K + Currents Shapes Perithreshold Resonance and Spike Tuning in CA1 Pyramidal Neurons. Jorge Vera 1, Julio Alcayaga 1, Magdalena Sanhueza. Front Cell Neurosci. 2017 Mar 8;11:61. doi: 10.3389/fncel.2017.00061.3. Intrinsic bursting activity in the pre-Bötzinger complex: role of persistent sodium and potassium currents. Ilya A Rybak 1, Natalia A Shevtsova, Krzysztof Ptak, Donald R McCrimmon. Biol Cybern 2004 Jan;90(1):59-74. doi: 10.1007/s00422-003-0447-1. Epub 2004 Jan 21.4. Persistent Sodium Current, Membrane Properties and Bursting Behavior of Pre-Bötzinger Complex Inspiratory Neurons in vitro Christopher A. Del Negro, Naohiro Koshiya*, Robert J. Butera Jr. and Jeffrey C. Smith 01 NOV 2002, https://doi.org/10.1152/jn.00081.2002.

We are grateful to the third reviewer who pointed out to us the three papers of possible relevance to our study (the last one by Vera et al., deals with the unrelated issue of membrane resonance), although these articles do not concern embryonic development.

However, there are major differences between these previous studies and our manuscript, and their relevance is quite limited:

1. The experimental study by Del Negro et al., shows that bursting occurs in neonatal inspiratory neurons of the Pre-Bötzinger complex when the ratio of the persistent sodium conductance (G_Nap_) to the leak conductance is large but the work does not analyze the role of the balance between sodium and potassium voltagedependent conductances as we did. It does not make much sense to consider this ratio in our study as the input conductance of embryonic Renshaw cells shows little spread at E12.5 and does not evolve between E12.5 and E14.5, at variance with the ratio of G_Nap_ to the delayed rectifier conductance G_Kdr_. Still, we analyzed the effect of the input conductance when explaining the results of blocking G_Kdr_ with 4-AP.

2. In the model of adult hippocampal pyramidal cells of Golomb et al., bursting does not arise from the slow inactivation of I_Nap_ as in our study but from the slow inactivation of the potassium current I_M_ that is present in their model but that we had no reason to incorporate in ours, as this current is not present in embryonic Renshaw cells.

3. In the simulation paper of Rybak et al., on mature Pre-Bötzinger, the bursting that occurs at the transition from quiescence to repetitive firing is, as in Del Negro et al. and Golomb et al., of the usual “square wave bursting” type, i.e. an alternation of spiking episodes and quiescent periods. Our model can also predict this classical bursting mode, however it has never been observed in embryonic Renshaw cells. These cells display another type of bursting known as “pseudo-plateau bursting”, in which the plateau is stable for the fast dynamics and coexists with an unstable limit cycle emerging at a subcritical Hopf bifurcation, just the opposite of the square wave bursting found in Rybak et al. Our model allowed us to explain why this peculiar bursting mode was seen in Renshaw cells. Therefore, our model may be considered as a very general framework explaining how a strong diversity of firing patterns may occur in embryonic neurons. We also examined the effect of the A-current and of channel noise, two features relevant to Renshaw cells and that were not considered by Rybak et al.

Finally, we must point out that we did not focus our analysis to the restricted situation of firing onset as in the three studies above. We also studied plateau states and the occurrence of “elliptic bursting” at the transition from repetitive firing to plateau, which is beyond the scope of the three articles mentioned by the reviewer. This was done through the mathematical analysis of the dynamics (bifurcation diagrams, slow/fast analysis), contrarily to Rybak et al., who relied on numerical simulations only. Such an analysis was feasible and realistic from a physiological point of view because we discovered that embryonic Renshaw cells express a limited number of voltage-gated channel subtypes but display a large repertoire of activity patterns, five different ones, including two unusual bursting modes. This is more diverse than the firing patterns studied in Del Negro et al., Golomb et al. or Rybak et al.

Therefore, our model can be considered as an original, general and realistic model explaining a wide variety of firing patterns observed experimentally, the transitions between these patterns, and it shows that the relevant control parameter is the ratio GNap/GKdr.

We added the mentioned three papers in the bibliography of our manuscript and also briefly discuss them now in the sub-section entitled “Theoretical analysis: slow inactivation of I_Nap_ and bursting” of the Results section and in the sub-section entitled “Ion channels mechanisms underlying the function heterogeneity of embryonic V_1_^R^” of the Discussion, emphasizing the differences with our model, the wider scope of our modified Hodgkin-Huxley-like model, and the originality of our results (See pages 17-18 lines 408-443 and pages 21-22 lines 517-545).